# Effect of Street Lighting on the Urban and Rural Night-Time Radiance and the Brightness of the Night Sky

Tomasz Ścięzor 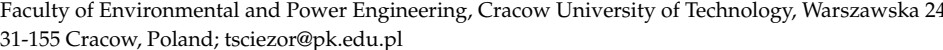

Faculty of Environmental and Power Engineering, Cracow University of Technology, Warszawska 24, 31-155 Cracow, Poland; tsciezor@pk.edu.pl

**Abstract:** In April 2020, due to the coronavirus pandemic and the tourism decrease in Cracow (Poland), the Road Authority of the City of Cracow, followed by the authorities of several neighbouring municipalities, decided to turn off street lighting at night. It is worth noting that this is the first time that street lighting has been turned off in such a large area on a longer time frame at this scale, including one of the most populated cities in Poland, which made it possible to make unique observations. During this period, with the help of small night-sky radiance meters (Sky Quality Meters (SQM)), many ground-based measurements were made, both within the city and in its vicinity. For this purpose, the existing light pollution monitoring stations in Cracow and neighbouring cities were used. It was found that after switching off street lighting, the zenith surface brightness of the cloudless sky decreased by 15–39%, and this value was proportional to the city's population. The night-time light satellite data (VIIRS/DNB) on radiance from Cracow and neighbouring communes were also analysed, both their daily values as well as monthly and annual averages. It was found that in the case of a large city such as Cracow, turning off all street lighting reduces the amount of light energy radiated into the sky by about 50%, which is a relatively small decrease in radiance, while reducing the surface brightness of the night sky by about 40%, regardless of the state of the atmosphere. The effect of a significant decrease in radiance as a result of switching off street lighting was found in each of the analysed communes, especially the urbanised ones.

**Keywords:** light pollution; remote sensing; radiance; street lightning; artificial sky glow

## 1. Introduction

From the very beginning of the problem of light pollution, especially in the form of an artificial night sky glow, it was obvious that the main sources of this problem were cities, particularly street lighting [1,2]. The concept of a light glow was first introduced probably in 1935 [3], and the concept of astronomical light pollution in 1973 [4]. In the following years, the described phenomenon increasingly threatened astronomical studies [5], including those carried out in Cracow (Poland) [6]. Along with the development of urban infrastructures, in addition to street lighting, other light sources appeared, increasing the brightness of the sky glow. Light pollution is mainly due to the operation of wrongly designed street lighting, advertising, illumination of sports stadiums, construction, security, and façade lighting [7–11]. To this day, there is no real effective policy that deals with this problem.

As a result of studies on the impact of urban lighting on the brightness of the night sky, several theoretical models were created to describe this problem [12–21].

The impact of cities on light pollution was also the subject of field studies, enabled by the introduction of cheap Sky Quality Meters (SQMs), which became a kind of measurement standard [22–25]. Similar measurements were also carried out in 2009 in Cracow and the surrounding area by the team of the Light Pollution Monitoring Laboratory (LPML) operating at the Cracow University of Technology, led by the author of this paper [26,27]. The obtained results showed compliance with the values predicted by the first World Atlas of the Artificial Night Sky Brightness [28].

From 1972, it became possible to quantify the light emission from cities based on satellite images taken by the Defence Meteorological Satellite Program (DMSP) satellites [29,30]. In 2001, using the Treanor and Garstang models and determining the amount of light emitted by cities based on DMSP images, the aforementioned atlas was created (often called the Atlas of Light Pollution), commonly used to describe light pollution on Earth's surface [28]. However, since the purpose of the DMSP satellites was to forecast the weather, their cameras did not have sufficient resolution to show the fine structure of light sources.

In 2011, the National Oceanic and Atmospheric Administration (NOAA) launched the Suomi NPP satellite [31]. This satellite has a visible infrared imaging radiometer suite camera (VIIRS) providing the best freely available night-time dataset with daily and global coverage. While in the case of DMSP satellites the smallest visible objects were 3 km in size, the VIIRS images show objects smaller than 1 km [32]. The fundamental importance of VIIRS data, in particular Day–Night Band (DNB) images for mapping Earth's surface night illumination, especially when compared to earlier DMSP data, was highlighted in 2013 [33]. VIIRS data were also used to create the all-sky artificial sky glow model [34] and the new World Atlas of the Artificial Night Sky Brightness [35].

In 2008, the Luojia 1-01 CubeSat (6U) sized earth observation satellite, built by the Wuhan University (China), was launched into orbit. This is the first dedicated night-time light satellite providing images with a spatial resolution of 130 m, which is much better than that of DMSP/OLS and Suomi-NPP/VIIRS [36]. This creates new perspectives for remote sensing of night lights [37].

Additionally, to analyse the impact of individual types of urban lighting, high-resolution photos of some cities taken from the International Space Station (ISS) can be used [38,39]. Such a photograph was also taken for Cracow on 28 March 2017 (Figure 1). In this photo, almost all types of illuminated streets can be distinguished, as well as separate parts of the city where metal-halide or high-pressure sodium lamps (HPS) have been replaced by LED lighting.

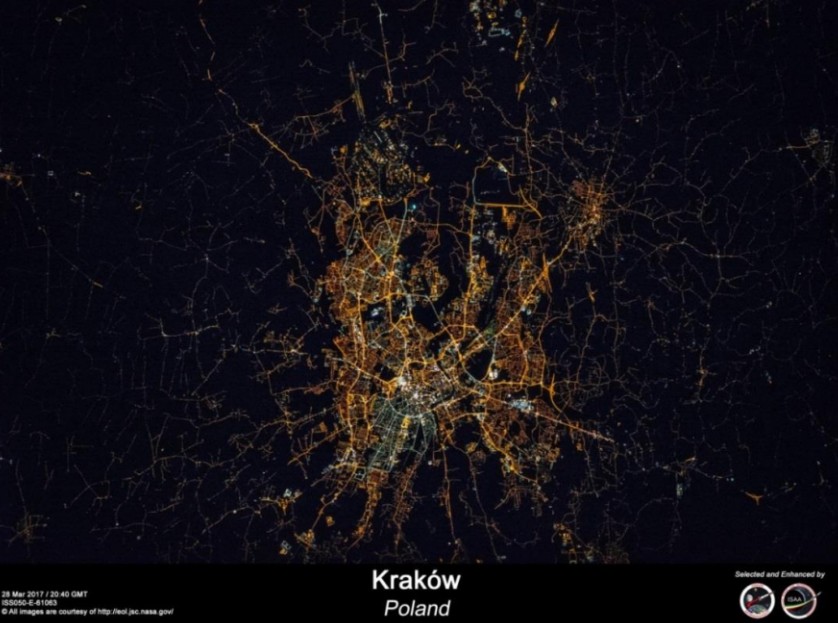

**Figure 1.** ISS photo of Cracow (Kraków) taken 28 March 2017 (Courtesy of http://eol.jsc.nasa.gov/, accessed on 22 April 2021).

One of the first analyses of the contribution of particular types of urban lighting to the city's total radiance was the 2009 estimation of the complete outdoor lighting of Flagstaff (Arizona) [40]. It was found that the average fraction escaping directly upward from light fixtures was equal to 8.3%. Of the total uplight, 33% came from sports lighting when it

is on; when sports lighting is off, commercial and industrial lighting accounted for 62%, with the remainder dominated by residential (14%) and roadway lighting (12%).

The potentially most complete data on light pollution sources could be obtained by switching them off sequentially and analysing the effect of these switching offs on the night sky glow. Thus far, city lighting outages have been short lived, related to lighting failures, and have occurred in a limited area, most often in a certain part of the city. However, these events were always an opportunity to research their effect on the brightness of artificial sky glow.

One of the first studies of this type was conducted on 28 September 2006 in Reykjavik, where all streetlights were shut off from 10:20 p.m. to 10:40 p.m. to prepare for the Reykjavik International Film Festival [41]. It was found that street lighting was a significant contributor to sky-glow light pollution.

Another example can be a power failure in part of Cracow in the late evening of 25 August 2011 [42]. During the failure, for more than two hours, a compact area of 2 km$^2$ with approx. 50,000 population was deprived of any power supply (both flats and street lighting). The sky brightness was then equal to 18.07 mag/arcsec$^2$ (6.44 mcd/m$^2$). When repairing the failure, only the closest vicinity of the measurement site, with an area of about 0.03 km$^2$, was deprived of light for about half an hour. The measured zenith surface brightness of the sky was then equal to 17.72 mag/arcsec$^2$ (8.90 mcd/m$^2$). After the failure was completely repaired, the zenith surface brightness of the virtually cloudless sky, measured near the centre of this region, was equal to 17.51 mag/arcsec$^2$ (10.79 mcd/m$^2$). Measurements performed during the phenomenon allowed for the analysis of the impact of local lighting on the artificial sky glow of the entire city. It was found that the complete switching off of lighting in an area constituting only 4% of the city's area causes a 40% reduction in the zenith surface brightness of the night, cloudless sky.

In 2016, differential photometric measurements of the change in night sky brightness and illuminance were performed during an automated regular switch-off of ornamental light in the town of Balaguer and an organised switch-off of all public lights in the village of Àger, both near Montsec Astronomical Park in Spain [43]. It was found that during the clear night, the ornamental lights in Balaguer contribute over 20% of the sky glow at zenith at the observational site, while very small changes were found in the ground illuminance on a cloudy night near Àger. Similar research, but on a local scale, was conducted in an urban park (Tiergarten) in Berlin (Germany) during the WWF Earth Hour in 2018 [44]. It was found that turning off nearby lighting (Potsdamer Platz) resulted in a 15% decrease in scalar illuminance, 11% decrease in horizontal illuminance, and 8% decrease in zenith luminance.

To find out what fraction of outdoor light emissions and associated energy use are due to streetlights or various types of private light sources, an experiment was carried out in 2019 in Tucson (Arizona). The streetlight output had been intentionally altered over 10 days, and the change in emissions observed by satellite was examined. It was found that streetlights operated by the city are responsible for only 13% of the total radiance measured from Tucson from space after midnight [45]. If the city did not dim its streetlights after midnight, the contribution would be 18%. When streetlights operated by other actors are included, the best estimates rise to 16% and 21%, respectively.

Another analysis of this experiment showed that the zenith sky brightness during the tests decreased by 5.40% near the city centre and 3.6% at an adjacent suburban location on nights when the output of the street lighting system was dimmed from 90% of its full power draw to 30% after local midnight [46]. The model of light sources, consisting of 26% known street lights and 74% of other sources results suggests that street lights account for about 14% of light emissions resulting in sky glow seen over the city. However, the direct measurements implied that street lighting contributed only 2–3% of light seen at the zenith over Tucson. The difference between the modelled and experimentally determined values was explained by an underestimation of the total city light emission, not accounting for the transient sources of light whose emission patterns are near horizontal, such as interior lighting escaping through uncovered building windows, automotive lighting,

and illuminated signs. The improvement of sky brightness may also be related to air pollution in the studied areas [47,48].

The spreading of a COVID-19 pandemic in 2020 influenced the increase in the burden on health care, decreased human mobility, and the related decline in tourism and caused various kinds of unexpected effects in the natural environment as well [49]. As a result of migration restrictions and industrial activity limitations, overall carbon emissions have dropped [50]. An improvement in air quality and a decrease in water pollution in many cities around the globe were also noticed [51]. The impact of the pandemic was even visible in aspects such as the reduction of seismic noise associated with anthropogenic earth surface vibrations from trains, aeroplanes, industrial processes, etc. [52].

Of course, the pandemic lockdown also prompted many light pollution researchers to investigate the changes it caused in the brightness of a sky glow. The effects of the lockdown on urban light emissions were described for Granada (Spain), including both ground and satellite data [47]. The research, conducted from 14 March until 31 May 2020, showed a clear decrease in light pollution due to both a decrease in light emissions from the city and a decrease in anthropogenic aerosol content in the atmosphere which resulted in less light being scattered. A clear correlation between the abundance of PM10 particles and sky brightness was observed, such that the more polluted the atmosphere, the brighter the urban night sky. The ground-based data showed little change in the second half of the night, compared to the first, and also the satellite imagery does not show marked differences during the lockdown in terms of total light output. Given that outdoor activities decreased by up to 90% during a lockdown, these results indicate that the light output of the city seems to be dominated by permanent lighting that does not adapt to the real use of the outdoor areas by the citizens.

A photometric mapping of the artificial sky glow of a large city and its surroundings was also performed on the example of Berlin, both in the case of a cloudless and completely overcast sky [48]. It was found that the brightness of the artificial sky glow at zenith decreased by 20% at the city centre and by more than 50% at a 58 km distance from this area during the lockdown. At the same time, the overall light emission trend in the region, as observed by satellite data, was increasing. It has been suggested that the main cause for the reduction of artificial sky glow originates from improved air quality due to less air and road traffic, which is supported by statistical data and satellite image analysis. A similar analysis is currently being carried out for Cracow by the author of this paper, to be published in the future.

In April 2020 and the following months, the authorities of Cracow, as well as several neighbouring communes, resolved to completely or partially turn off street lighting at night, starting from midnight (or from 1:00 CEST) to dawn (usually to 4:00 CEST). This created a unique opportunity to study the contribution of street lighting to both the emission of light energy from the entire city (especially since VIIRS/DNB images of the research area are taken just after 1:00 CEST, i.e., after turning off street lighting in all researched communes), as well as to measure changes in the surface brightness of the night sky. A network of automatic and manual LPML measuring stations was used for these measurements. The analysis of both these data sets is the subject of this paper.

## 2. Materials and Methods

### 2.1. Research Area

The research was conducted in the western part of Małopolskie Voivodeship (Figure 2). The Małopolskie Voivodeship consists of 182 communes, including 14 urban communes (also the City of Cracow, constituting a separate urban commune), 48 urban–rural communes, and 120 rural communes [53]. Based on press releases, information on switching off the lighting in Cracow and 20 neighbouring communes was analysed. Only in eight communes (Alwernia, Chrzanów, Cracow, Dobczyce, Krzeszowice, Skawina, Wieliczka, and Zabierzów) in 2020 street lighting was actually turned off for at least one month

(Appendix A: Table A1). There is one urban commune (Cracow) and one rural commune (Zabierzów), and the rest are urban–rural communes with varying degrees of urbanisation.

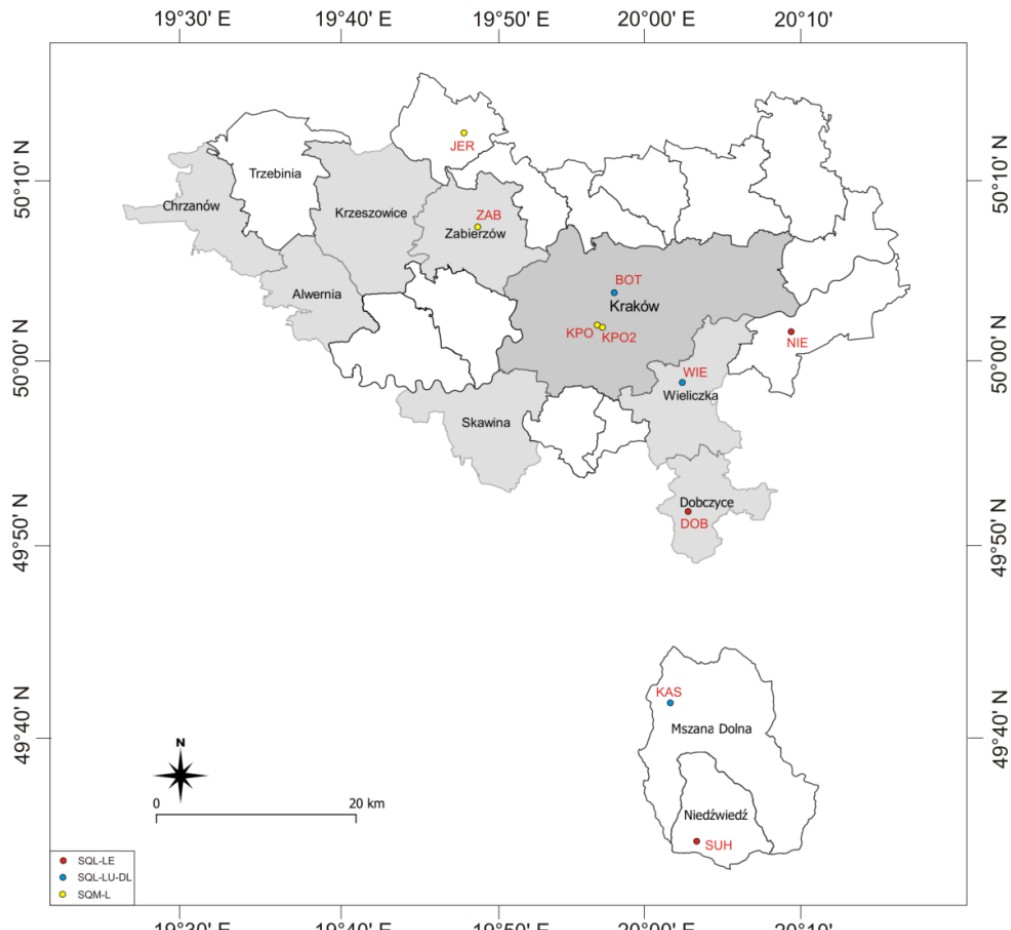

**Figure 2.** Map of the western part of Malopolskie Voivodeship showing the city of Cracow (dark-grey shading) along with the surrounding communes. The communes in which street lighting was turned off in 2020 are marked grey. The commune of Trzebinia is also featured, where street lighting was not turned off in 2020; hence it is treated as a control area in the analysis of radiance changes. To the south of the research area, the communes of Mszana Dolna and Niedźwiedź are marked, where control measurements of the brightness of the night sky were performed during the research. The dots indicate the location of night-sky brightness measuring points (red: automatic Ethernet SQM-LE, blue: automatic data logging SQM-LU-DL, yellow: handy SQM-L).

2.1.1. Road System

Data on the length and the type of roads were obtained from the GIS Open Street Map (OSM) for all selected communes. According to the road classification system, the following categories are distinguished in Poland:

- Motorway: these are at least two-lane highways, divided by a median, with each direction lit separately. In the research area, the motorways are mainly the ring roads of Cracow, practically illuminated along their entire length (CIE M Class);
- Primary: the most common are national two-lane high-speed roads, also illuminated along their entire length (CIE M Class);
- Secondary: most often provincial two-lane roads. They are connections between large cities. They are also illuminated along their entire length (CIE M Class);
- Tertiary: usually the paved county roads. They connect cities that are the seats of poviats (the second-level unit of local government and administration in Poland) with the seats of communes and seats of communes with each other. In cities, they are

the main inter-residential streets, often also used by public transport. They are also two lanes, illuminated along their entire length (CIE M Class);

- Unclassified: in cities, they are supplementary to tertiary streets; outside cities they are communal roads. Most often illuminated on one side (CIE M Class);
- Residential: usually streets in cities and generally accessible roads in built-up areas for local access to properties or small settlements. One-sidedly lit (CIE P Class);
- Service: access roads, e.g., to parking lots or estate buildings; service roads, in-house transport, property driveways, etc. Usually poorly lit by single lamps (CIE P Class);
- Living street: streets in residential zones. A zone in which pedestrians can move freely throughout the public space available for public use and has priority over vehicles. Usually dimly lit by low-power lamps (CIE P Class);
- Pedestrian: streets accessible only to pedestrians, e.g., in the old towns. Usually dimly lit by low-power lamps (CIE P Class);
- Footway: pedestrian paths, sidewalks. Usually lit independently of adjacent roadways (CIE P Class);
- Cycleway: bicycle paths. Usually using the lighting of adjacent roadways or sidewalks (CIE P Class).

In each of the analysed communes, the length of roads from each category (of course, excluding unlit paths, field roads, etc.) were measured taking into account the information which parts of its lighting were turned off in 2020 (Appendix B: Table A2).

Figure 3 shows a map of illuminated roads in Cracow compared to the corresponding mean monthly VIIRS/DNB image for March 2020 [33,54]. Similar maps for all other analysed communes are provided in Appendix C (Figures A1 and A2). The course of primary and secondary roads is visible in the VIIRS/DNB images.

### 2.1.2. Street Lighting Shutdowns

In April 2020, the authorities of the city of Cracow (Kraków), the capital of the Małopolskie Voivodeship, decided to turn off street lighting at night throughout the city. This decision was made in connection with the decline in traffic with the growing COVID-19 coronavirus pandemic. Following Cracow, the authorities of several neighbouring communes also made a similar decision to a varying extent.

The lighting was turned off automatically at designated times. This process was controlled by astronomical clocks placed in lighting control cabinets in various lighting zones. Due to the inaccurate timing of individual clocks, the times of switching off the lighting in various areas of a given commune could differ by up to 10 min (which was actually observed in, e.g., Cracow). The process of tuning all clocks in the commune sometimes took several days.

The action of switching off city lighting in Cracow had the widest range [55]. On 8 April, the lighting in parks was completely turned off. From midnight on 15 April, street lighting within the city limits, subordinate to the Cracow Roads Authority, was completely turned off. Street lighting was turned off at midnight and turned back on at 4:00 CEST. Since 22 April, the lighting of parks has been synchronised with street lighting. From 19 May, the switching off of street lighting in Cracow was moved to 1:00 CEST, and it was shut down until dawn (about 4:00 CEST). From midnight on 1 June, standard lighting was restored throughout the city.

At the time of switching off the street lighting in Cracow (Kraków), the only active sources of artificial lighting were lamps located on private properties, lamps subordinate to housing communities, and various types of advertising lighting (Figure 4).

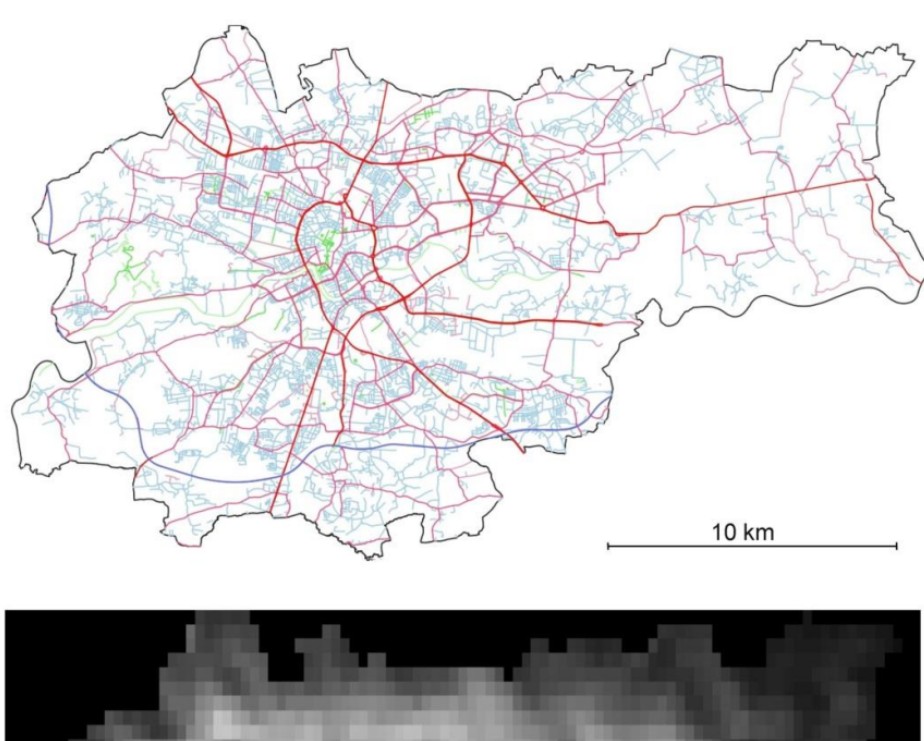

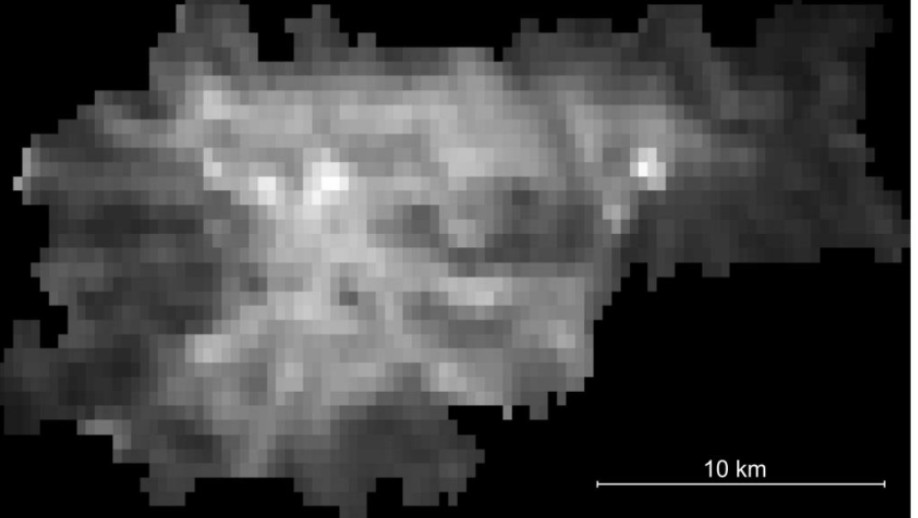

**Figure 3.** The GIS road map of Cracow (upper picture). The colours correspond to the OSM classification: navy blue—motorway; shades of red—primary, secondary, tertiary, and unclassified; blue—residential and service; green—footway; yellow—cycleway. Below is the monthly VIIRS/DNB Cloud Mask image generated for March 2020, before the street lighting shutdown period.

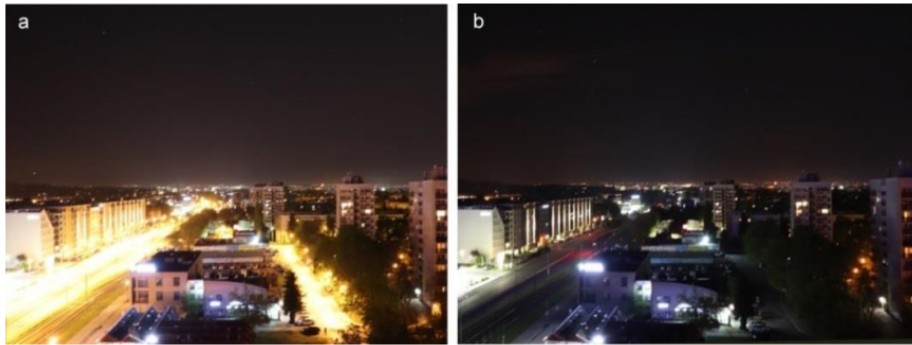

**Figure 4.** Turning off street lighting in Cracow. Pictures were taken with the same Canon EOS 800D camera settings (ISO 1600, F 5.6, 3 s) about 20 min before midnight (**a**) and 15 min after midnight (**b**) on the night of 26–27 April 2020. The only constantly glowing light sources are the petrol station (on the left), some neighbourhood lamps, apartment windows, security lamps, and light advertisements (Photo: Tomasz Ścieżor).

The following neighbouring communes followed the example of Cracow (Figure 2):

- In the Alwernia commune, switching off the street lighting began on 19 May 2020. Between 23:30 and 4:00 CEST, the lighting of all streets in the city, subordinated to the commune authorities, except for the 10 main streets, was turned off [56]. This lighting has been restored in September, but in November, the lighting of roads was turned off in the entire area of the commune between midnight and 4:30 CET. This state lasted until February 2021 [57].
- In the Chrzanów commune, switching off the street lighting, subordinate to the commune authorities, began on 25 June 2020. A total of 4518 lighting points with a power of 599,590 W, connected to the power grid in 143 energy consumption points, were turned off from midnight to 4:00 CEST. Lighting was restored in September 2020 [58].
- In the Dobczyce commune, from 20 May 2020, street lamps were switched off in some rural places with low car traffic. The lighting was turned off from midnight to 5:00 CEST. Starting in November, some lighting circuits have reduced the shutdown time to 4:00 CEST. These activities are still ongoing (February 2021) [59].
- In the Krzeszowice commune, switching off the street lighting began on 23 April. From midnight to 4:00 CEST, the lighting of all streets, subordinate to the commune authorities, was turned off. Street lighting was gradually restored in July 2020, but in September, it was turned off again, mainly in the rural areas of the commune. Standard lighting was restored in December 2020 [60].
- In the Skawina commune, gradual switching off the street lighting from midnight to 4:00 CEST lasted from 13 April to 18 May in various parts of it. The lighting of all streets, subordinate to the commune authorities, as well as in the City Park and housing estates, was turned off finally in June. Night lighting was gradually restored from 1 September to 14 September 2020 [61].
- In the Wieliczka commune, switching off the street lighting began on 17 April, and the process was gradually continued in the following days. As planned, street lighting was turned off at midnight and turned on again at 4:00 CEST. However, due to the fact that some of the clocks were still operating according to CET (thus they turned off the lighting at 1:00 CEST) and required correction, it was only in May 2020 that the lighting of all streets subordinate to the commune authorities was turned off from midnight. Standard lighting in the Wieliczka commune was restored on 1 September 2020 [62].
- In the Zabierzów commune, switching off the street lighting started in mid-May. From 1 June, the lighting of all streets subordinate to the commune authorities was turned off from midnight to 4:00 CEST. National, voivodeship, and poviat roads as well as some sections of communal roads connected to them remained illuminated. Standard lighting in the Zabierzów commune was not restored until 1 January 2021. It should be noted that a railway line runs through the centre of the commune, which has its own lighting, not under the control of the commune authorities. There is also a very brightly lit greenhouse complex, which is not subject to the commune authorities [63].

### 2.1.3. SQM Measuring Points

The research area covered the city of Cracow with the surrounding communes. In some of them (Figure 2, marked with a grey background), in 2020, street lighting was turned off at night, most often between midnight and 4:00 CEST/CET. For all these communes, both VIIRS/DNB data analysis and SQM measurements of the brightness of the night sky were conducted. For this purpose were used both manual SQM-L and automatic continuous measurement SQM-LE and SQM-LU-DL, owned by LPML. Unfortunately, the pandemic conditions and the related constraints caused that these data are sometimes incomplete. The reasons were either difficulties in the movement of people (in the case of SQM-L) or difficulties in accessing stationary autonomous meters for the replacement of exhausted batteries (in the case of SQM-LU-DL). Moreover, the unexpected announcement of pandemic restrictions resulted in a delay in starting the new measuring points or running temporarily disabled ones.



Measurements with SQM meters were conducted in communes where street lighting was turned off (Cracow, Wieliczka, Zabierzów, Dobczyce), in other neighbouring communes (Jerzmanowice, Niepołomice), and at measuring points away from the research area, treated as control (Kasinka Mała, Mt. Suhora) (Figure 2) (Table 1; details in Appendix D).

**Table 1.** The SQM measuring points. $S_{a\ max}$ means the greatest value measured during the research.

| Name | Lat (N) | Long (E) | Alt [m] | SQM Type | $S_{a\ max}$ [mag/arcsec$^2$] |
|------|---------|----------|---------|----------|-------------------------------|
| BOT | 50°03′50″ | 19°57′22″ | 209 | SQM-LU-DL | 18.8 |
| DOB | 49°52′17″ | 20°04′23″ | 273 | SQM-LE | 21.2 |
| JER | 50°07′12″ | 19°46′58″ | 238 | SQM-L | 20.6 |
| KAS | 49°41′30″ | 20°02′19″ | 390 | SQM-LU-DL | 21.4 |
| KPO | 50°02′23″ | 19°55′35″ | 202 | SQM-L | 18.8 |
| KPO2 | 50°02′23″ | 19°55′49″ | 202 | SQM-L | 18.8 |
| NIE | 50°02′17″ | 20°12′40″ | 196 | SQM-LE | 20.3 |
| SUH | 49°34′09″ | 20°04′02″ | 1009 | SQM-LE | 21.7 |
| WIE | 49°59′25″ | 20°04′16″ | 253 | SQM-LU-DL | 20.0 |
| ZAB | 50°07′12″ | 19°46′58″ | 238 | SQM-L | 20.2 |

## 2.2. Remote Sensing

In order to determine the total radiance from individual communes, photos obtained from the Suomi National Polar-orbiting Partnership (SNPP) satellite were used. This satellite was launched in 2011 by the US National Oceanic and Atmospheric Administration (NOAA) [31]. The satellite has a Visible Infrared Imaging Radiometer Suite (VIIRS) camera equipped with the best sensors and optical systems in civilian missions so far. While in the case of the earlier Defence Meteorological Satellite Program (DMSP) satellites the smallest visible objects were 3 km in size, the VIIRS images show objects smaller than 1 km in size.

For this research, images taken in the Day–Night Band spectral range (DNB) [64] are useful. This range has a wavelength centre at 700 nm, a bandpass of 400 nm, and a resolution of 0.8 km. DNB is, in a sense, a special sensor within VIIRS. It has separate detectors, a different imaging method, but above all a higher sensitivity. DNB operates in three signal amplification modes. Therefore, in DNB, at night, much more details can be observed than in "day" bands. Raw DNB data require careful treatment and consideration of many factors, including clouds or the angle of the photo. Therefore, the analysis used previously processed data.

When analysing monthly mean radiance, GeoTIFF files obtained from Earth Observation Group [33,54] were used, in particular monthly cloud-free DNB composites contained in vcmsl files. These files include all data, also impacted by straight light, but which underwent the stray-light correction procedure.

To perform a comparative analysis of the radiance for specific nights, before and after the time of switching off the street lighting in a given commune, the daily VIIRS/DNB data were also analysed, taken from Level-1 and Atmosphere Archive and Distribution System Distributed Active Archive Center at Goddard Space Flight Center (LAADS DAAC) [65]. The first of two VIIRS DNB-based datasets is a daily, top-of-atmosphere, at-sensor night-time radiance product called VIIRS/NPP Daily Gridded Day Night Band 15 arc-second Linear Lat Lon Grid Night. This product, called VNP46A1, contains 26 science sata sets (SDSs) that include sensor radiance, zenith and azimuth angles (at sensor, solar, and lunar), cloud-mask flags, time, shortwave IR radiance, brightness temperatures, VIIRS quality flags, moon phase angle, and moon illumination fraction. It also provides quality flag (QF) information specific to the cloud mask, VIIRS moderate-resolution bands M10, M11, M12, M13, M15, M16, and, what is important for this analysis, the DNB. During the research period (March 2020 to January 2021) VIIRS/DNB images of the research area were

taken around 0:10 UTC, i.e., 1:10 CET or 2:10 CEST (depending on the season). Since the astronomical clocks that turned off the street lighting were working according to CEST (summer) or CET (autumn and winter), VIIRS/DNB images had already been taken when the lighting was turned off. It should be noted that the elimination of sunlit data due to the high latitudes of the areas of research results in a lack of data near the summer solstice. As a result, it is not possible to analyse the daily data in June 2020.

For determining the average annual radiance, the VIIRS/DNB data obtained via the Light Pollution Map website [32] were used.

It should be noted that the paper describes changes in the total radiance in the analysed areas not the radiance per pixel.

Used Units

From the VIIRS/DNB data, the radiance is obtained, i.e., power per unit projected area per unit solid angle. The symbol of radiance is L, and its standard SI unit is $W/m^2$ sr [66]. In this paper, radiance values are given in the derived units $nW/cm^2 \cdot$ sr, commonly used in the VIIRS/DNB data.

### 2.3. SQM Measurements

Measurements were made using Sky Quality Meters (SQMs) by Unihedron (Appendix E). SQMs are produced in several versions, of which in this research were used manually triggered SQM-L, computer-controlled SQM-LE, and autonomous SQM-LU-DL. All these meters have a light gathering angle of 20° from the instrument axis. All meters, while taking measurements, were aimed at the zenith with an accuracy of approx. 3°.

Used Units

The SQM gives the brightness of the night sky in the astronomical unit of magnitude per square arcsecond ($mag/arcsec^2$) (Appendix E). By assuming a linear correlation between the perceived surface brightness (luminance) and the physical surface brightness, a relationship can be obtained [67–69] as follows:

$$(mag/arcsec^2) = 12.59 - 2.5 \log[(cd/m^2)] \tag{1}$$

or vice versa,

$$[cd/m^2] = 10.8 \cdot 10^4 \cdot 10[-0.4(mag/arcsec^2)] \tag{2}$$

Of course, these conversions are only approximations because of the different spectral responses of SQM and the human eye [70]. In this publication, the SQM measurement results are given in $mag/arcsec^2$, but when performing the analysis, especially when comparing with VIIRS/DNB data, they are converted into $mcd/m^2$.

Multiple measurements have shown that the amplitude of changes in the $S_a$ value in each tested meter under stable atmospheric conditions does not exceed 0.02 $mag/arcsec^2$. Therefore, for the SQM-L, the measurement had been triggered until three consecutive readings were equal with an accuracy of 0.01–0.02 $mag/arcsec^2$, and no changing trend was observed. As a result, 0.02 $mag/arcsec^2$ was assumed as the short-term repetition error.

In the case of SQM-LE and SQM-LU-DL meters, the analysis included only the data obtained during the period when street lighting was switched off; $S_a$ changes did not exceed 0.03 $mag/arcsec^2$. This value is assumed as the short-term measurement accuracy.

The long-term analysis of the SQM measurements (e.g., when comparing results from different nights) should take into account its quoted systematic uncertainty of 10 per cent (0.1 $mag/arcsec^2$) [71].

### 2.4. Information on Street Lighting Shutdowns

Information on the spatial and temporal range of street lighting shutdowns in 2020 was obtained from the authorities of all communes where such activities were undertaken. In addition, information was also received in the form of a questionnaire from the inhab-

itants of the communes of Alwernia, Jerzmanowice-Przeginia, Krzeszowice, Wieliczka, Niepołomice, and Zabierzów (see Section 2.1.2).

## 3. Results

### *3.1. Remote Sensing Data*

All VIIRS/DNB images are presented in the EPSG coordinate system: 4326—WGS 84/Spherical Mercator (Pseudo Mercator).

### 3.1.1. The Commune and City of Cracow

To check whether the average annual radiance from the area of Cracow changed in 2020, compared to the previous years, its value and other statistical parameters were examined for 2012–2020 within the administrative borders of Cracow, covering the area of 325.86 km$^2$ (Table 2). In this area, the statistics for 2365 pixels were determined. The variability of the mean annual radiance is shown in Figure 5. The increase in radiance in 2017 related to the modernisation and replacement of approximately 3000 sodium lamps with LEDs is noticeable both in Table 2 and in Figure 5.

**Table 2.** Mean annual radiance in Cracow in 2012–2020 [nW/cm$^2$·sr] [32].

| Year | 2012 | 2013 | 2014 | 2015 | 2016 | 2017 | 2018 | 2019 | 2020 |
|---|---|---|---|---|---|---|---|---|---|
| Sum | 45,145 | 48,208 | 44,924 | 44,168 | 45,319 | 49,360 | 47,378 | 47,673 | 46,439 |
| Mean | 19 | 20 | 19 | 20 | 19 | 21 | 20 | 20 | 20 |
| Std. dev. * | 17 | 17 | 17 | 17 | 17 | 19 | 17 | 17 | 17 |
| Min. | 1 | 1 | 1 | 0 | 1 | 1 | 1 | 1 | 1 |
| Max. | 87 | 121 | 116 | 109 | 101 | 148 | 113 | 133 | 124 |

*—Standard deviation (SD) of pixel values completely inside the area.

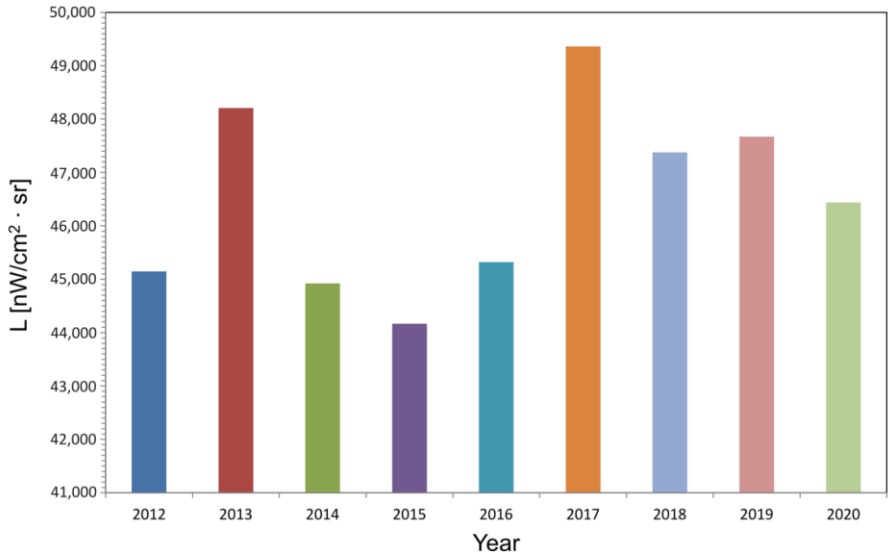

**Figure 5.** Mean annual radiance in Cracow in 2012–2020. In 2014–2015, 4375 street lamps were replaced. In 2017, approximately 3000 sodium street lamps were replaced with LEDs. In the following two years, the city lighting was modernised.

The mean monthly radiance of the area of research was also determined for the subsequent months of the year from March to July in the period 2015–2020 (Table 3 and Figure 6). January and February were skipped due to a large number of cloudy nights and thus lower reliability of the data. Very low radiance in May 2020, compared to all other months in the analysed period, can be noticed.

**Table 3.** Mean monthly radiance in Cracow from March to July in 2015–2020 [nW/cm$^2$·sr] (SD = 15–19) [33,54]. Bold font indicates the decrease in radiance in May 2020. Radiation changes in March and April should probably be associated with rain and snowfall in these months, changing the ground reflectance.

| Year | 2015 | 2016 | 2017 | 2018 | 2019 | 2020 |
|------|------|------|------|------|------|------|
| March | 48,514 | 44,609 | 50,118 | 49,848 | 46,980 | 51,472 |
| April | 49,512 | 53,845 | 45,060 | 51,766 | 46,457 | 44,258 |
| May | 50,408 | 46,421 | 46,917 | 48,187 | 41,316 | **26,311** |
| June | 44,073 | 47,593 | 49,045 | 48,842 | 55,552 | 46,661 |
| July | 42,403 | 45,116 | 45,769 | 42,390 | 47,659 | 46,547 |

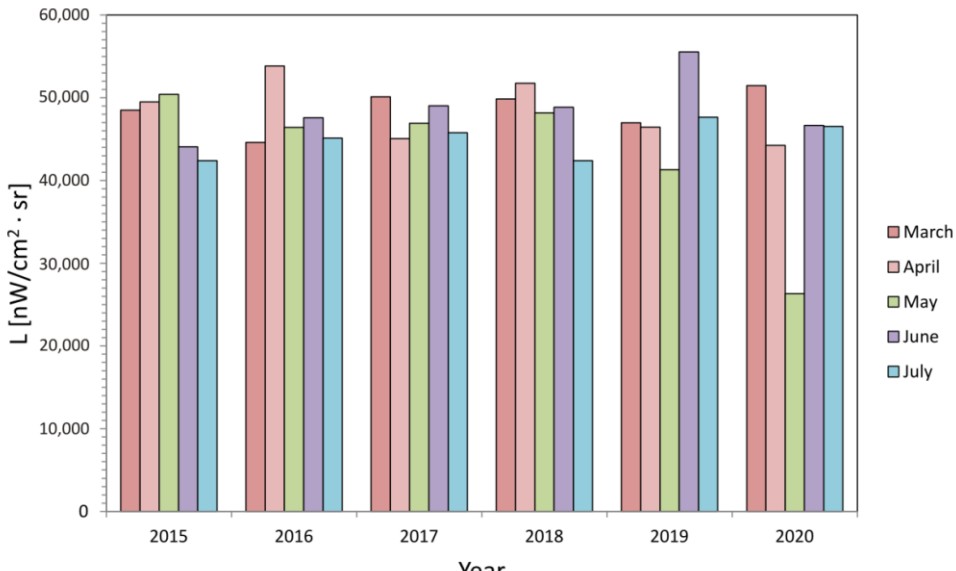

**Figure 6.** Mean monthly radiance in Cracow in 2015–2020. There is a clear decrease in radiance in May 2020.

Figure 7 shows the VIIRS/DNB images of Cracow in 2020, in March (on the left, before turning off the street lighting) and May (on the right, street lighting turned off for the whole month). To demonstrate what changes have occurred between these months, the differential radiance between March and May is shown below—all the bright pixels mark areas where the radiance has decreased.

To verify the available data on the dates of turning off and turning on the street lighting, the daily radiances for selected nights were analysed. To minimise the impact of other factors not related to switching off the street lighting, in particular clouds or reflected moonlight, the analysis was performed for the selected moonless and completely cloudless nights (Table 4).

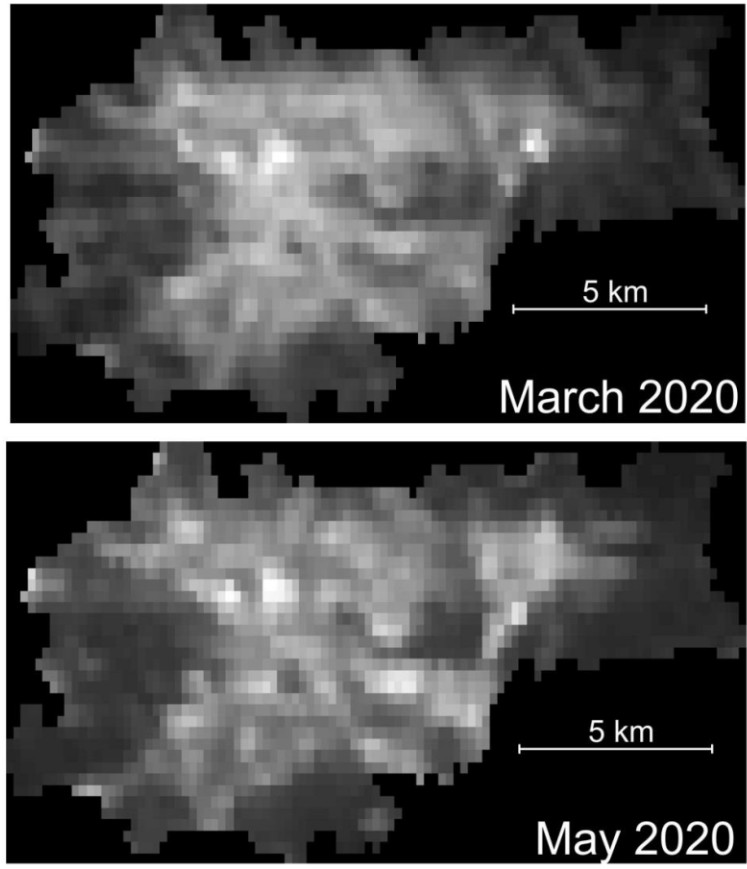

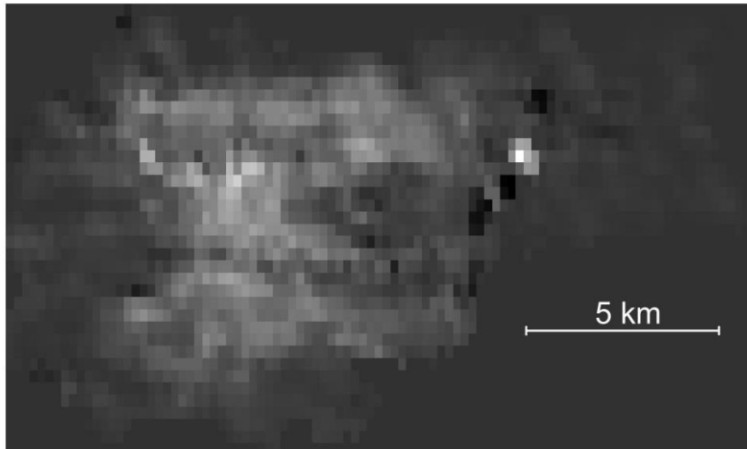

**Figure 7.** Mean monthly radiance of Cracow in March and May 2020. Below is the differential radiance between these months (VIIRS/DNB)—bright pixels mark areas where the radiance has decreased.

**Table 4.** Daily radiance for Cracow. Nights with street lighting completely turned off are shown in bold font, nights during which only part of the street lighting was turned off are shown in italic font (SD = 10–20) [65]. There is a noticeable reduction in radiance during the partial shutdown of street lighting in April 2020 and the complete shutdown of street lighting in May 2020.

| Date | Mean Daily Radiance [nW/cm²·sr] | Reason for Choosing the Date |
|---|---|---|
| 25.03.2020 | 55,986 | "normal" state of street lighting |
| 27.03.2020 | 53,090 | "normal" state of street lighting |
| 13.04.2020 | 52,468 | lighting of city parks turned off |
| *16.04.2020* | 33,063 | first day of switching off street lighting throughout the city |
| **4.05.2020** | 26,140 | switching off all street lighting throughout the city |
| **21.05.2020** | 29,477 | switching off all street lighting throughout the city |
| 19.07.2020 | 41,344 | vacation, after the complete restoration of street lighting |
| 20.08.2020 | 39,377 | vacation, after the complete restoration of street lighting |
| 23.09.2020 | 51,258 | "normal" state of street lighting |
| 23.10.2020 | 47,192 | "normal" state of street lighting |

### 3.1.2. Other Communes in the Małopolskie Voivodeship

For the aforementioned communes, the average annual radiances for the years 2012–2020 were also determined (Figure 8) (Appendix F: Table A3).

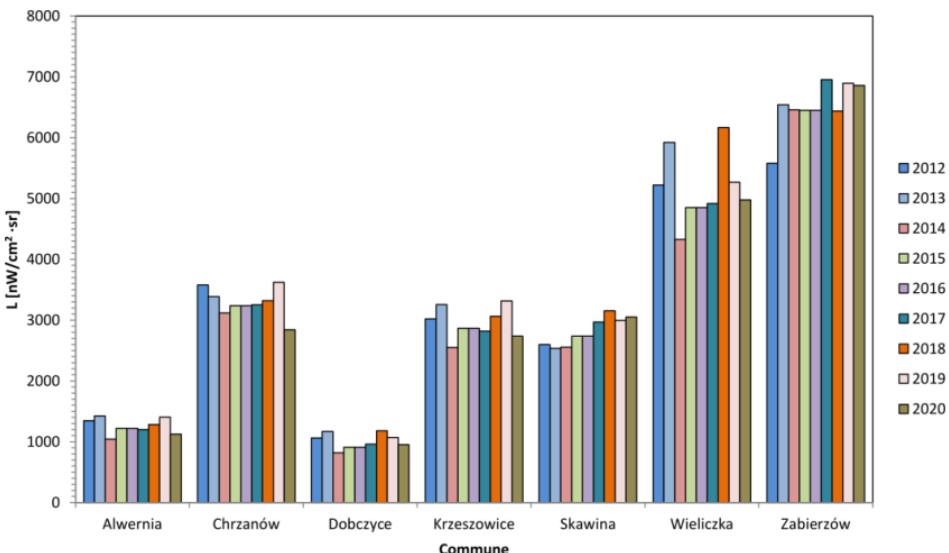

**Figure 8.** Mean annual radiances in near-Cracow communes in 2012–2020. A decrease in the radiance in 2020 is visible for the communes of Alwernia, Chrzanów, Dobczyce, Krzeszowice and Wieliczka. The effect of street lighting modernisation in some communes in 2017–2019 is also visible.

For the same communes, the mean monthly radiances in 2020 were also measured (Figure 9) (Appendix F: Table A4; see also Figure A3 for 2019, for comparison).

In almost every case, the decrease in radiance in the months when street lighting was turned off is evident. To exclude other possible causes of this phenomenon, the monthly radiances for the control communes (in which street lighting was not turned off) are also presented. For this purpose, the commune of Trzebinia was selected since it is a typical urban–rural commune, located between similar communes of Chrzanów, Alwernia, and Krzeszowice, where street lighting was turned off. The communes of Mszana Dolna

and Niedźwiedź have also been added since they are home to SQM measuring points, respectively, KAS and SUH. Additionally, these communes are located in the mountains far from communes where street lighting was turned off. A noticeable increase in radiance from all these test communes during the shutdown of street lighting in Cracow perhaps should be connected to the pandemic return of students from a large academic centre, such as Cracow, as well as the "pandemic escape" from this city some of its citizens who have their summer houses in these communes.

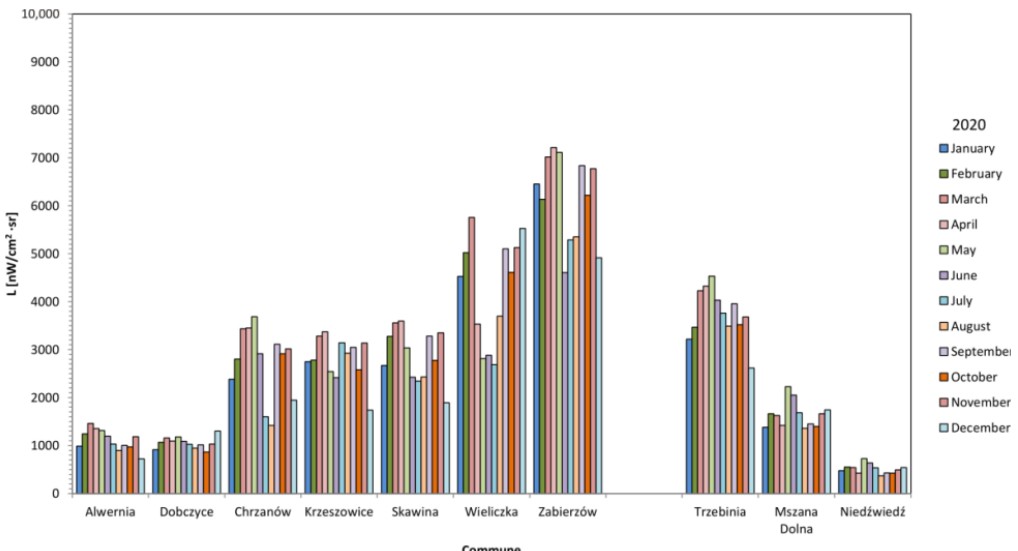

**Figure 9.** Mean monthly radiances in seven near-Cracow communes, in which street lighting was turned off, in the following months of 2020. In each case, a decrease in radiance is noticeable during periods when street lighting was off. Monthly radiances for control communes (Trzebinia, Mszana Dolna, Niedźwiedź), in which street lighting was not turned off, are also shown.

The mean change in monthly radiance in each of the analysed communes is summarised in Table 5. Differential radiance means the sum of the differences in the radiance from individual pixels in the analysed area between months immediately preceding the switching off period and the ones in which the greatest number of street lamps was switched off. Relative differential radiance is the quotient of the former value and the radiance from the period preceding lighting off.

An analysis of the daily radiance for the examined communes was also conducted. This analysis allows for determining how long the switching off or switching on process was carried out. To eliminate the factors that could potentially disturb the reported values, attempts were made to choose moonless, cloudless, or only slightly cloudy nights. For urban–rural communes, the components of radiance from urban and rural areas were also determined separately (Appendix F: Table A5, Figure A4).

Figure 10 shows the VIIRS/DNB images of the analysed communes for the nights with all street lamps lighting (at left) and for the nights when the lighting was completely or partially turned off (in the middle). To show what changes have occurred between these nights, the differential radiance between them is also shown (at right)—bright pixels mark areas where radiance has decreased. Figure 11 shows the changes in daily radiance from March 2020 to January 2021 for a series of nights in the discussed communes. Efforts were made to represent each month for at least one night (usually, in each commune street lighting was completely turned off at the beginning of a month). Only in June, due to the proximity of the summer solstice, it was impossible to find the appropriate VIIRS/DNP image.

**Table 5.** Change in the mean monthly radiance related to switching off the street lighting in the analysed communes (including Cracow). The used VIIRS/DNB data come from the months immediately preceding the switching off period and the ones in which the greatest number of street lamps was switched off. Both the absolute and relative percentage changes in radiance are given.

| Commune | Differential Radiance dL [nW/cm$^2$ · sr] | Relative Differential Radiance [%] |
|---|---|---|
| Alwernia | 417 | 29% |
| Chrzanów | 1798 | 49% |
| Dobczyce | 93 | 8% |
| Cracow | 25,162 | 49% |
| Krzeszowice | 956 | 28% |
| Skawina | 1171 | 33% |
| Wieliczka | 2941 | 51% |
| Zabierzów | 516 | 7% |

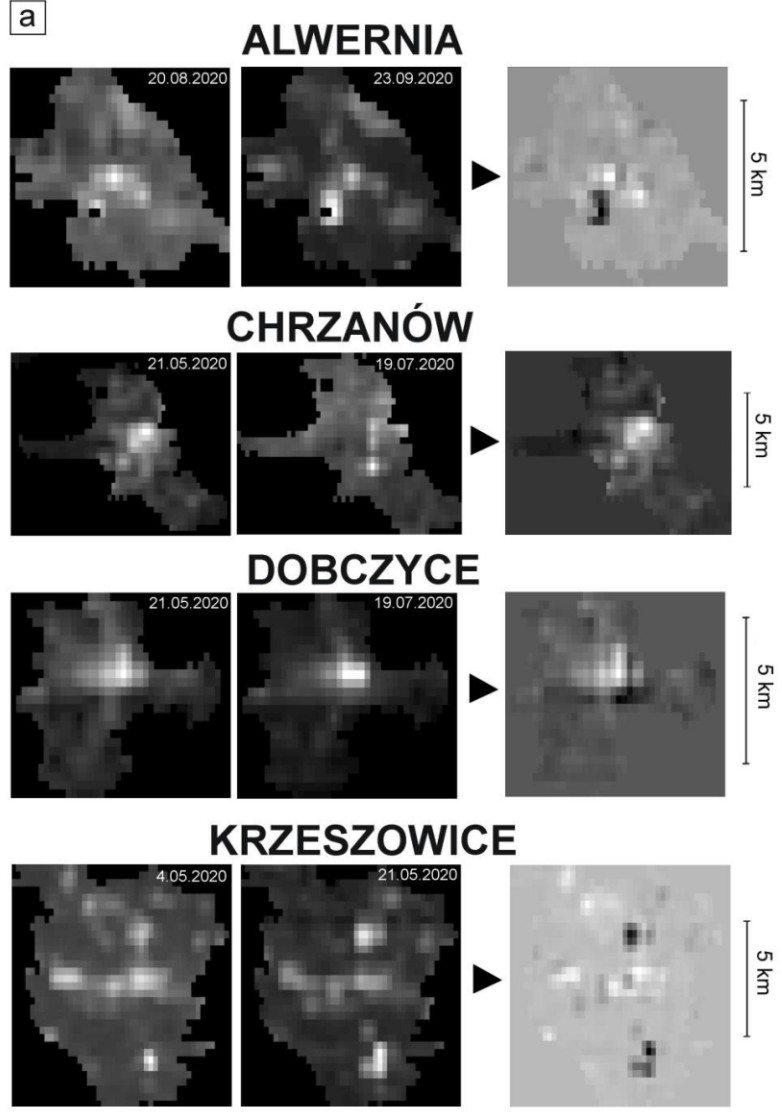

**Figure 10.** *Cont*.



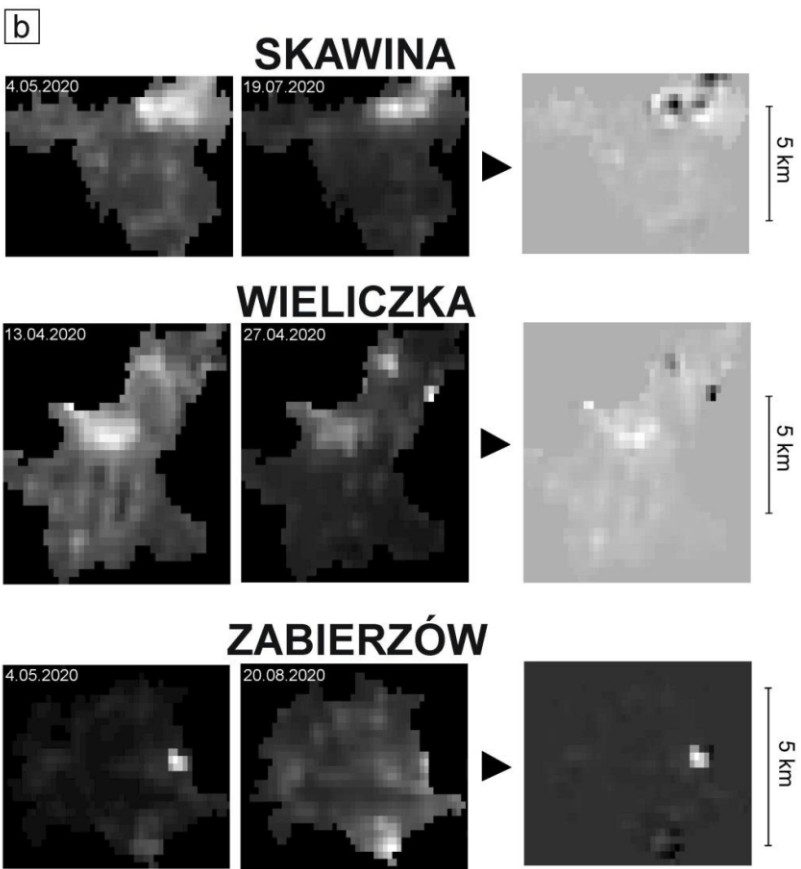

**Figure 10.** VIIRS/DNP images of the analysed communes (**a**—Alwernia, Chrzanów, Dobczyce, Krzeszowice; **b**—Skawina, Wieliczka, Zabierzów) on selected nights of 2020, before and after switching off the street lighting (left and middle column, respectively). To show what changes have occurred between these nights, the right column shows the differential radiance between the respective nights—bright pixels mark areas where radiance has decreased due to street lighting being turned off (the large greenhouse in Zabierzów is particularly outstanding).

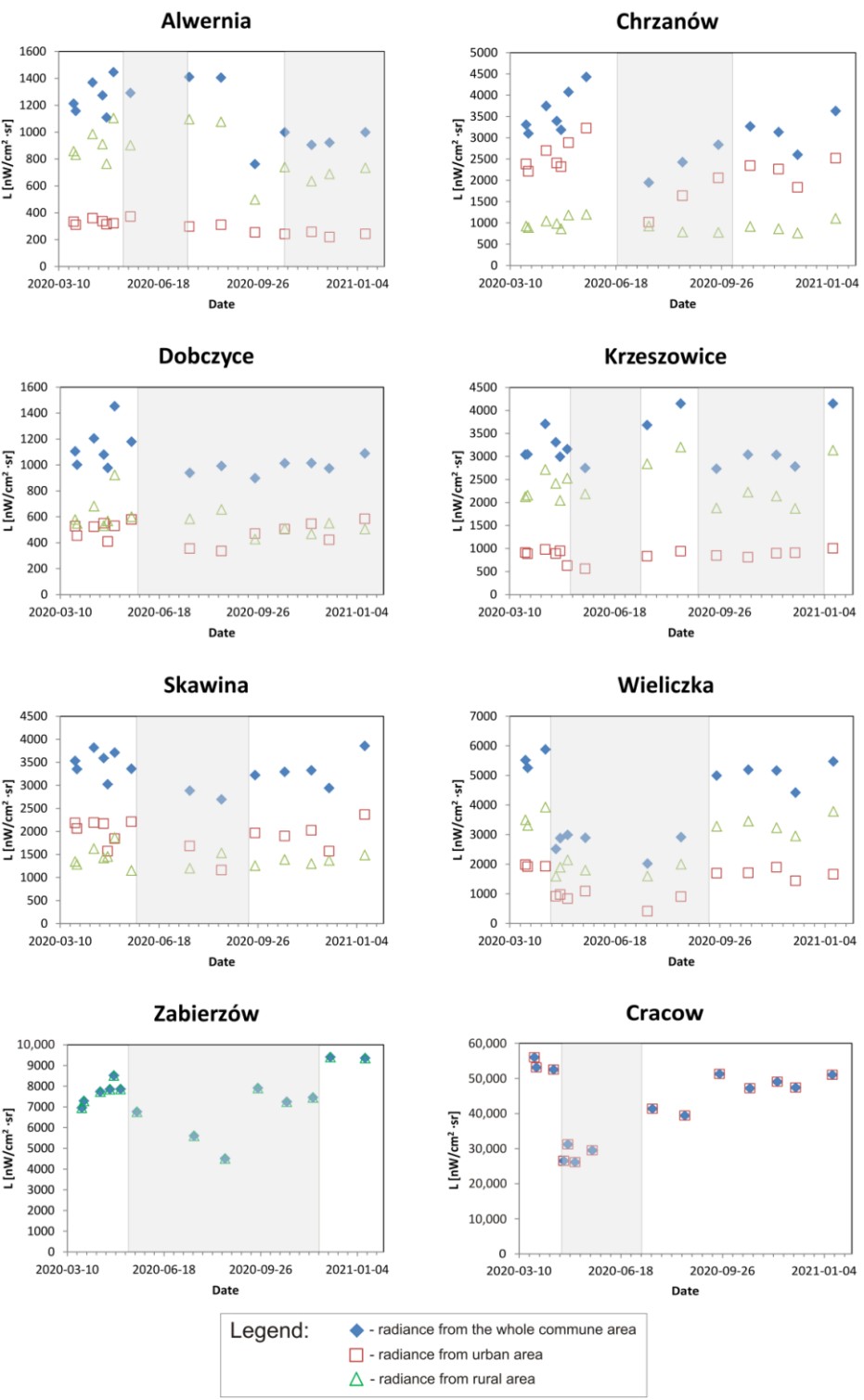

**Figure 11.** Changes in radiance in 2020 for the analysed communes of the Małopolskie Voivodeship (full diamonds: radiance from the whole commune area; empty squares: radiance from urban area; empty triangles: radiance from rural area). In each case, the dates of decrease/increase of radiance coincide with the dates of switching off/on street lighting, provided by municipal offices (marked with grey zones) [55–63].

*3.2. SQM Measurements' Data*

3.2.1. Measurements on Cloudless and Moonless Nights

With the SQM meters, a number of measurements of the brightness of the night sky were performed in the conditions of cloudless and moonless nights (Appendix G: Table A6). The changes in the mean value of $S_a$ measured at the measuring points were determined (Appendix G: Table A7).

During the entire measurement period, at the KAS (commune of Mszana Dolna) and SUH (commune of Niedźwiedź) control measuring points, no changes in the $S_a$ values were found that could be related to street lighting shutdowns in Cracow or the neighbouring communes. In the case of the KAS measuring point, the maximum measured value of $S_a$ was $21.25 \pm 0.01$ mag/arcsec$^2$, in the case of the SUH station, $21.78 \pm 0.01$ mag/arcsec$^2$.

Figure 12 shows examples of changes in $S_a$ value during selected nights for the measuring points where the continuous measurements were carried out: BOT, DOB, NIE, and WIE. On diagrams, the suggested sources of reducing the brightness of the sky glow at a given measuring point are indicated by an arrow.

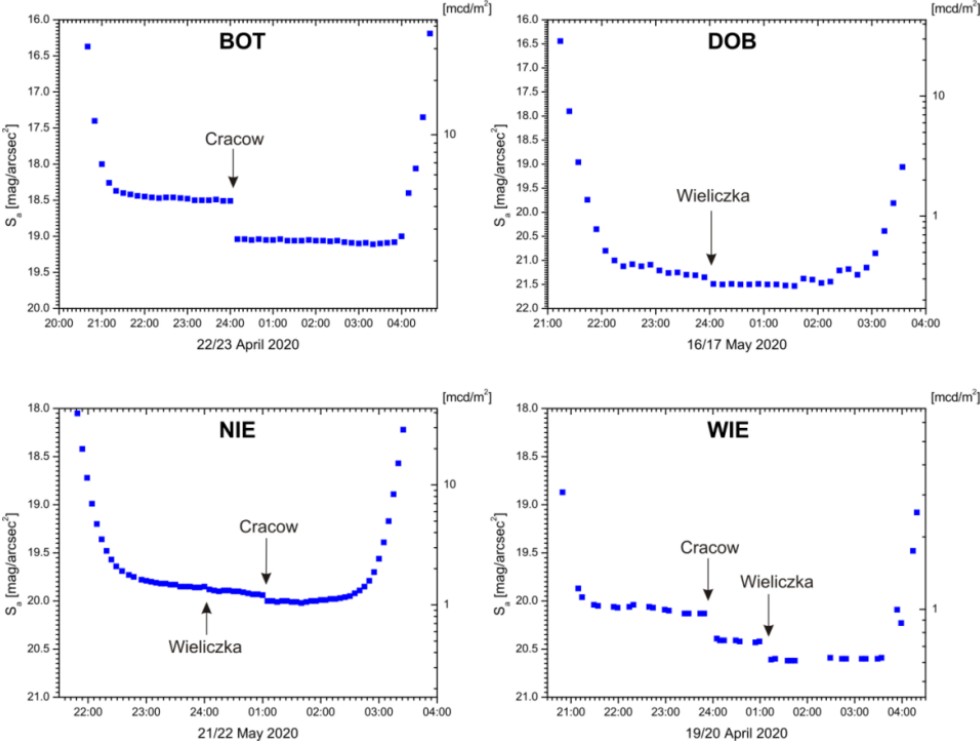

**Figure 12.** Examples of changes in the $S_a$ value during selected clear-sky nights for measuring points where the continuous measurements were carried out. The identification of sources of the reduction in the brightness of the sky glow in a given point is based on obtained official data and information from residents.

3.2.2. Measurements on Overcast Nights

Measurements were also carried out during cloudy nights (Appendix G: Table A8). Only at the measuring points KPO, KPO2, and ZAB the genus of clouds was recorded at the time of measurement. In the case of automatic measuring points (BOT, NIE, and DOB), the amplitude of $S_a$ variability during a cloudy night usually exceeded its possible changes related to the street lighting switching off; therefore, only some data from these points are possible for analysis (Figures 13 and 14).

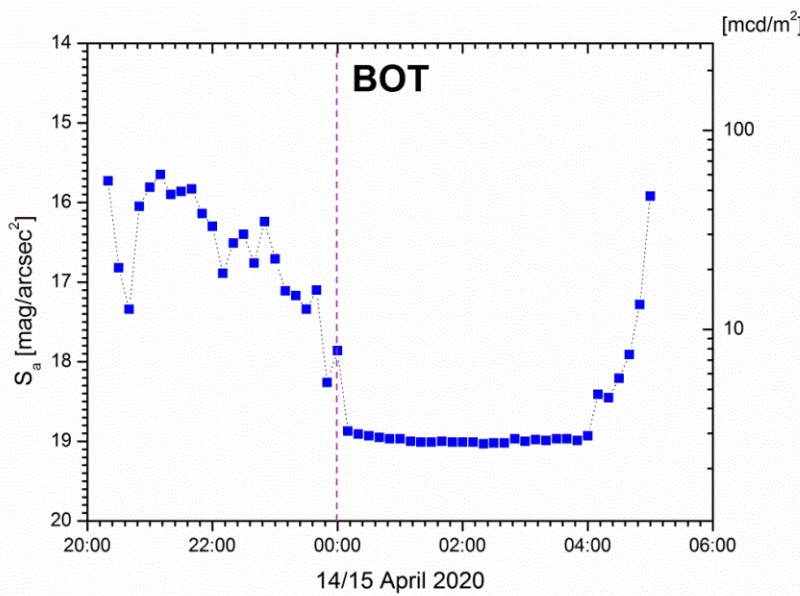

**Figure 13.** Changes in the $S_a$ value in Cracow (BOT measuring point) during the selected night (the first night of switching off street lighting in Cracow) with the sky overcast with middle- and high-level clouds. The dashed line indicates the time of switching off street lighting in Cracow.

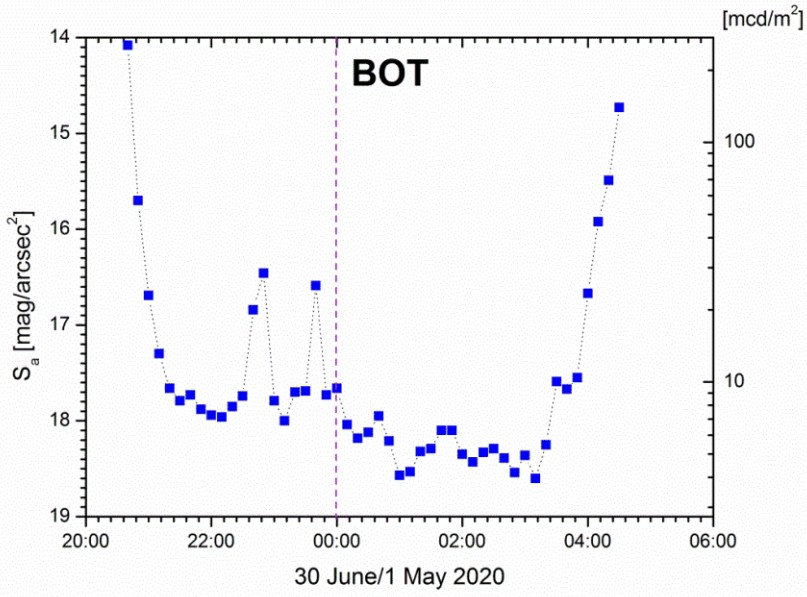

**Figure 14.** Changes in the $S_a$ value in Cracow (BOT measuring point) during the selected night with low-level clouds (Stratus). The dashed line indicates the time of switching off street lighting in Cracow.

## 4. Discussion

The number of illuminated roads in the commune, and particularly in the city, should depend on the number of inhabitants. Figure 15 shows the mean monthly radiance determined in March (before the street lighting shutting off period) vs. the population of the researched communes. There is a clear proportionality between these values. It means, as expected, that the main factor responsible for the amount of light energy emitted into the space is related to the number of emitters, which in turn is related to the population of the area. This relationship is consistent with the models proposed by Walker [14] and Berry [16].

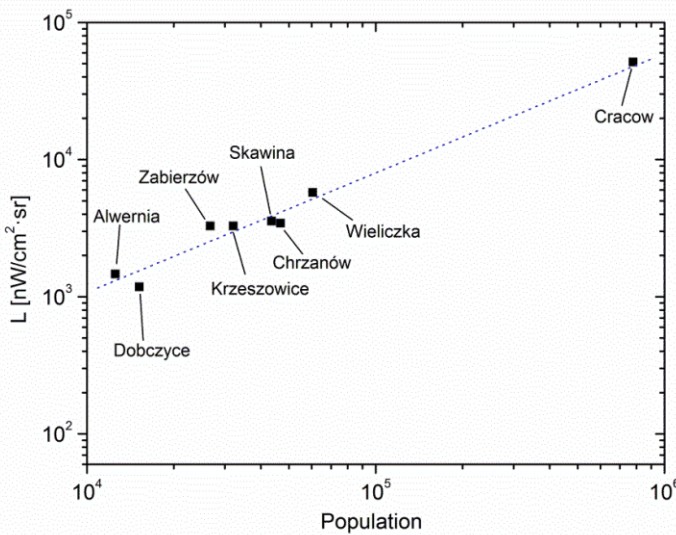

**Figure 15.** The mean monthly radiance vs. the population for the researched communes in the Małopolskie Voivodeship.

The presented research concerns the impact of switching off street lighting on the radiance. Therefore, the relationship between the radiance and the total length of illuminated roads in a given commune was checked. In this case, a clear proportionality of these values is also visible (Figure 16). However, it should be kept in mind that this relationship is indirect since undoubtedly the total length of roads in a given area depends on the population. Nevertheless, the presented graph means that unquestionably turning off street lighting should reduce the radiance. It can be concluded that the average radiance per kilometre of road in the research areas is 11.1 $nW/cm^2 \cdot sr/km$.

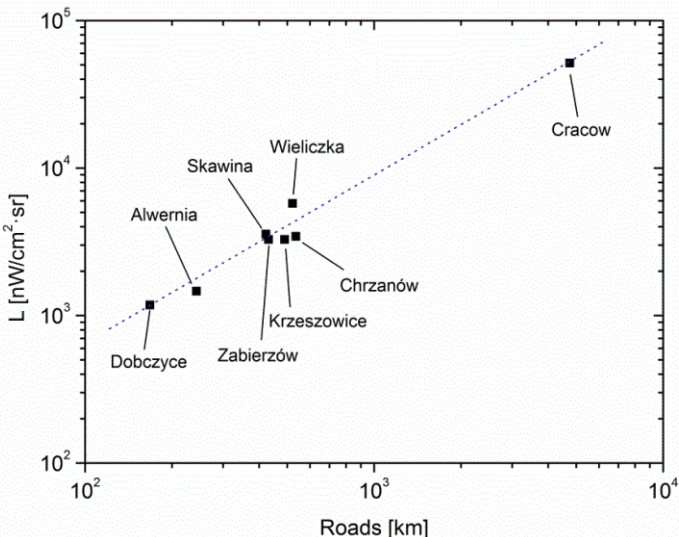

**Figure 16.** The mean monthly radiance vs. the length of illuminated roads in the analysed communes in Małopolskie Voivodeship.

Switching off street lighting in the analysed communes should impact their mean annual radiance. The column chart of the average annual radiance for Cracow in recent years (Figure 5) shows that the street lighting switching off in 2020 decreased this value, compared to the three previous years (in 2017, street lighting was modernised and extended in Cracow). To exclude the impact of other random factors that could cause such an effect, the monthly mean values of radiance for Cracow in the spring months of 2015–2020 were analysed (Figure 6). In May 2020, the radiance of Cracow significantly decreased, compared

to both the neighbouring months and the same month in the preceding years. This should, of course, be linked to the total shutdown of street lighting in the city for the entire month. In this way, it was shown that indeed switching off street lighting has a noticeable effect on the radiance. The obtained image of the differential radiance between March 2020 (when the lighting was not turned off) and May 2020 (Figure 7) shows that the effect of reducing the radiance was mainly caused by turning off street lighting in the western part of Cracow, mainly in the area of the Old Town and its surroundings (the lighting of the ArcelorMittal Poland steel plant in the eastern part of the city was also turned off). The analysis of the daily radiance on selected days in the research period (Table 4) shows that initially (when only part of the lighting was turned off), it decreased by approx. 19,400 nW /cm$^2 \cdot$ sr. At the beginning of May, on the days when the lighting of the greatest number of streets was turned off, the reduction in radiance compared to March was approx. 26,300 nW/cm$^2 \cdot$ sr, which means a reduction in the radiance of Cracow by half. This value is much higher than in the case of tests carried out with the street lighting switched off in Tucson (46). The reason is probably that Tucson has long had a very well-designed street light system that complies with protection against light pollution, while in Cracow, the modernisation of street lighting began only in 2017.

Analogous diagrams made for near-Cracow communes showed that the type of commune and the scale of switching off street lighting are of significant importance here. The effect of reducing the mean annual radiance in 2020, compared to previous years, is most clearly visible in the case of Alwernia (280 nW/cm$^2 \cdot$ sr), Chrzanów (780 nW/cm$^2 \cdot$ sr), Krzeszowice (580 nW/cm$^2 \cdot$ sr), and Wieliczka (290 nW/cm$^2 \cdot$ sr), i.e., communes in which a significant part of street lighting was turned off, or this turning-out lasted several months (Appendix F: Table A3). This effect is not visible in the case of Skawina and Zabierzów. The effect of switching off street lighting is particularly evident in diagrams showing the mean monthly radiance (Figure 9). The period of radiance reduction in Chrzanów, Krzeszowice, Skawina, or Wieliczka is fully consistent with the period of street lighting switching off (Appendix F: Table A4). However, the rapid decrease in radiance in the case of Zabierzów is rather not related to switching off street lighting but to the annual temporary turning off the lighting of the aforementioned greenhouse in summer months. In the case of Dobczyce, only part of the street lighting in some rural areas was turned off, which caused only a slight change in the commune's radiance.

Confirmations of the dates of switching off and switching on the street lighting are the daily radiances of individual communes (Figure 11; Appendix F: Table A5).

The analysis of VIIRS/DNB images for the dates preceding the periods of switching off the street lighting and for the periods during which it took place allows, to some extent, to identify the main light sources (Figure 3; Appendix C: Figures A1 and A2). The differential radiance images are also helpful in the identification. On their basis, it can be concluded that the radiance changes were indeed caused by switching off the lighting of the main roads in a given commune. However, turning off street lighting in towns and larger cities was of particular importance, which is especially well visible in the case of Alwernia, Chrzanów, and Wieliczka. In the case of Zabierzów, it was possible to find in the database the night during which the bright greenhouse, usually dominant in the image of this commune, was turned off—it is clear that it is indeed the main source of light in the commune. To identify the sources of radiance changes in urban–rural communes more confidently, it was determined separately for urban and rural areas (Figure 11; Appendix F: Table A5, Figure A4). In the case of poorly urbanised communes, such as Alwernia or Krzeszowice, radiance changes are caused only by turning off the street lighting in rural areas, which is evident especially for September 2020, when the lighting of rural roads was turned off for savings. In the case of urbanised communes, such as Chrzanów or Skawina, radiance changes are only caused by switching off the city lighting (in Chrzanów, the process of the gradual restoration of street lighting in August 2020 is visible). In the commune of Dobczyce, only part of the street lighting in rural areas was turned off. Wieliczka is the only commune where street lighting has been consistently

turned off, both in the city and in rural areas. In the case of the rural commune of Zabierzów, the observed reduction in radiance is less due to switching off the street lighting and to a much greater extent switching off the lighting of the aforementioned greenhouse in July 2020. In the case of Cracow, which is a municipal commune, naturally, switching off the street lighting in the city is responsible for the entire radiance change. A detailed territorial distribution of radiance and the linking of its changes with specific objects in the city is the subject of a separate publication that is being prepared.

Information was obtained from the analysed communes about which roads' lighting was turned off (Appendix B: Table A2). It is confirmed by the analysis of radiance images and their comparison with the road maps (Figure 3; Appendix C: Figures A1 and A2). These data make it possible to plot the relationship between the differential radiance of communes (the difference of radiance between the month in which street lighting was switched on and that in which street lighting was switched off) and the length of roads in which lighting was switched off (Figure 17).

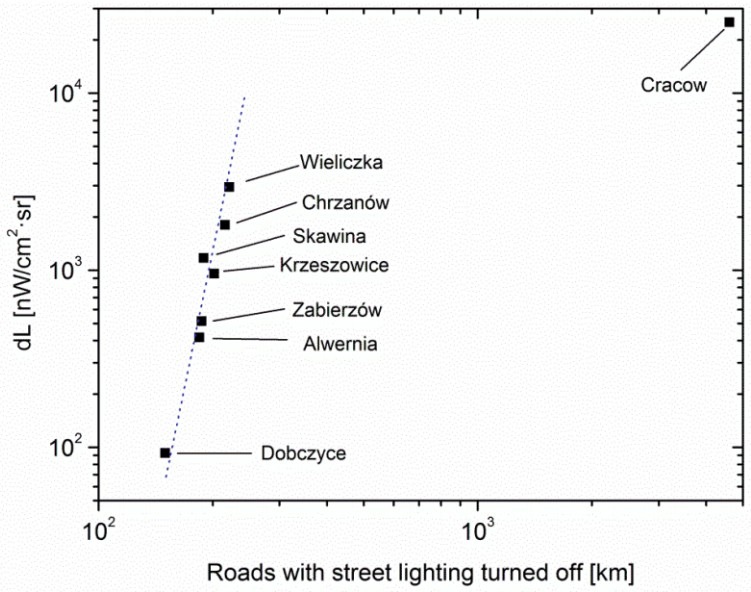

**Figure 17.** The mean monthly differential radiance in the analysed communes vs. the total length of roads in which lighting was turned off (in the case of the Zabierzów commune, a bright greenhouse was not included).

In the case of rural and urban–rural communes, the proportionality of the differential radiance to the length of "darkened" streets is evident. In these surroundings, street lighting is the dominant factor emitting light into space. In the case of Cracow, where all public street lighting was turned off (both in more and less urbanised areas), this resulted in a relatively small decrease in radiance. Perhaps it is related to the fact that in Cracow it is not the dominant factor illuminating the night sky. Additional sources of light are numerous apartment communities with their own lighting, as well as light advertising, offices, and even some important road intersections. It is also possible that this effect is caused by a higher concentration of particulate matter, which is the source of light scattering.

In communes, where the $S_a$ measuring points were active (Dobczyce (DOB); Cracow (KPO, KPO2, BOT); Wieliczka (WIE)), it is possible to plot a relationship between the decrease of $S_a$ and the decrease of radiance (Figure 18). Unfortunately, the small number of these measuring points makes it difficult to conduct a deeper analysis. However, it seems that the decrease in the surface zenith brightness of the sky ($dS_a$) depends mainly on the population of the commune not on the decrease in radiance. In the case of measuring points located in Cracow, $dS_a$ is equal to 0.5–0.6 mag/arcsec$^2$ (37% decrease), while in the case of neighbouring small cities, it is equal to 0.2–0.3 mag/arcsec$^2$ (17% decrease). In the case of Cracow, this value is much higher than the zenith brightness decrease in the

Tucson experiment [46]. The reason is probably that street lighting in Cracow began to be modernised only in 2017, and this modernisation is still underway.

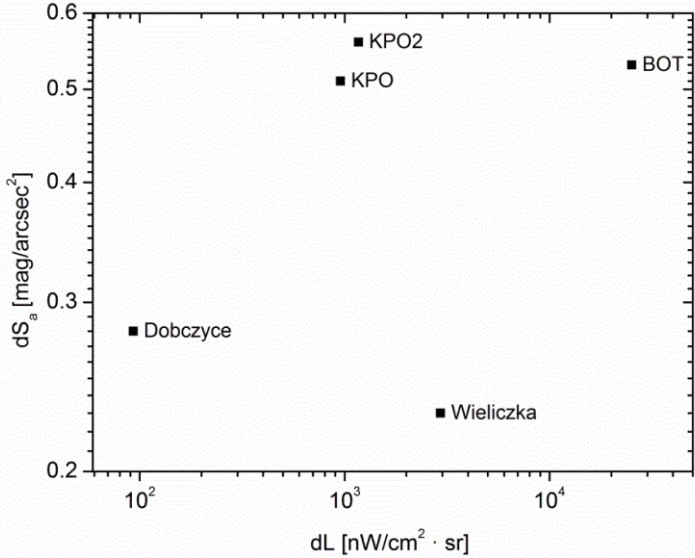

**Figure 18.** The change in the zenith surface brightness of the cloudless sky ($dS_a$) for the measuring points located in Cracow (BOT, KPO, KPO2) and in the Dobczyce and Wieliczka communes vs. the change in radiance (dL) of these communes.

At several measuring points, located outside Cracow, a decrease of $S_a$ can be related to the turning off the street lighting in this city. Figure 19 shows the relationship between the $S_a$ value measured at all active measuring points and the distance from the centre of Cracow, defined as the centre of the area with the greatest radiance change (i.e., the Old Town, Figure 7). A clear relationship between these values is visible. Moreover, as can be observed, even at a distance of 20 km, the changes of street lighting in Cracow affect the zenith brightness at this measuring point. A similar decrease in zenith brightness with distance was also found for Berlin [48], but in the case of Cracow, the range is much smaller. This should be associated with the much smaller population of Cracow (almost 800,000), compared to the population of Berlin (over 3,500,000).

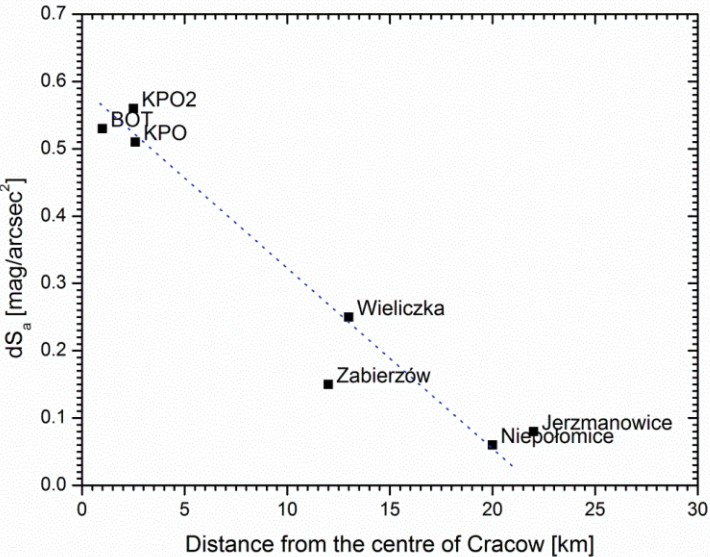

**Figure 19.** Change in zenith surface brightness of the cloudless night sky associated with switching off street lighting in Cracow vs. the distance from the centre of Cracow.

A similar analysis, but with a smaller number of cases, can be carried out on the example of the Wieliczka commune. Additionally, in this case, switching off the street lighting in the city impacts the zenith surface brightness of the night sky at the measuring points located in the neighbouring communes, up to a distance of approx. 12 km (Figure 20).

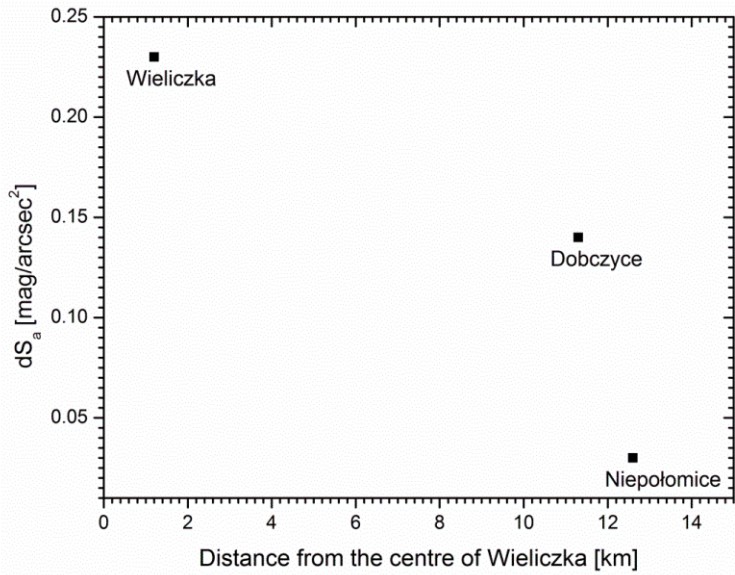

**Figure 20.** The change in the zenith surface brightness of the cloudless night sky associated with switching off street lighting in the Wieliczka commune vs. the distance from the centre of Wieliczka.

No change in the value of $S_a$, which could be associated with the shutdowns of street lighting in Cracow or the surrounding communes, was recorded at the measuring points in Kasinka Wielka (KAS, 40 km from Cracow) and at Suhora Mt. (SUH, 55 km from Cracow).

The above considerations apply to the case of the cloudless sky. However, at several measuring points, it was possible to analyse the impact of switching off street lighting in Cracow on the surface brightness of the overcast sky.

The most complete research of this type was carried out at the BOT measuring point located in the centre of Cracow. To determine the degree of cloudiness, and especially the genus of clouds, both the method of discrete derivatives [72,73] and the data of the meteorological station located at this point can be used. Throughout several nights, it was observed that in the presence of high-level clouds (and few middle-level clouds), when street lighting was turned off, the surface brightness curve typical for a cloudy sky "flattened", assuming the appearance characteristic of a cloudless sky (Figure 13). This is in line with previous observations [73,74], according to which the noticeable scattering of light on the high-level clouds occurs only in large cities with strong radiance. A similar effect was not observed in other cities where the brightness of the night sky was continuously measured (Wieliczka and Dobczyce). It is also not visible, even in Cracow, in the case of low-level clouds, e.g., of the Stratus genus (Figure 14).

At the KPO and KPO2 measuring points, located within Cracow, where manual measurements of the sky surface brightness were carried out along with noting the genus of clouds and the degree of cloud cover, it was found that with the low-level clouds, the $S_a$ value at the moment of switching off the street-lighting changes by 0.5–0.7 mag/arcsec$^2$, and this value does not depend on the genus or altitude of clouds. Analogous measurements carried out in Zabierzów (ZAB) and Jerzmanowice (JER) at the moment of switching off street lighting in Cracow showed that with low-level clouds of the genus of Stratus and Nimbostratus, the effect of changing $S_a$ is very weak and amounts to only approx. 0.06 mag/arcsec$^2$, while in the case of Stratocumulus clouds, it approaches 1.00 mag/arcsec$^2$. This means that both the altitude of clouds and their genus may be important. Unfortunately, a small number of measurements does not allow for more detailed

conclusions. However, it seems that these observations are consistent with those reported in the literature [74,75].

## 5. Conclusions

The analysis of VIIRS/DNB data showed that in each of the researched communes switching off street lighting significantly decreased the radiance. This effect is most clearly visible in urban or urban–rural communes with a high degree of urbanisation. For these communes, including Cracow as a commune, as a result of switching off all street lighting in cities, the radiance decreased by approx. 50% of values before the pandemic. In the case of rural communes (or communes with a dominant rural function), the corresponding decrease in radiance was approx. 35%. This means that in cities half of the light energy emitted into the sky comes from badly constructed street lighting. The other half comes from the lighting of private properties, parking lots, illuminated advertisements, etc. In communes with a predominant rural function, about one-third of the light energy is lost in this way—the remaining two-thirds come mainly from the lighting of individual farms and industrial plants. Unfortunately, the research covered only one rural commune (Zabierzów), where the dominant contribution to radiance comes from a single, bright object—a large greenhouse. Switching off its lighting reduced the commune's radiance by approx. 45% of the value before the pandemic.

Turning off street lighting and the resulting reduction in the amount of energy emitted to the sky obviously reduce the brightness of the sky glow, both in the case of cloudless and cloudy skies. In the case of a cloudless sky, the difference between the decrease in the surface brightness of the sky glow over the large city (Cracow) (amounting to approx. 1.7 mcd/m$^2$, i.e., approx. 39%) and the same value determined for smaller cities, such as Dobczyce and Wieliczka (amounting to 0.4 and 0.2 mcd/m$^2$, i.e., 22% and 15%, respectively), is evident. It is interesting that turning off all street lighting in Cracow reduces the radiance by, in fact, such a small amount. This effect and the difference between the relative decrease of radiance from Cracow and the one from smaller cities should probably be associated with the various concentration of atmospheric light scattering centres, especially with the density of particulate matter (PM). The decisive impact of PM on the brightness of the cloudless sky has already been found earlier by the author of this paper (it should be kept in mind that Cracow is one of the most polluted cities in the world) [27,76]. The research by Jechow and Hölker [48] also indicates the possibility of such correlation in the pandemic period in Berlin, but a similar effect in the Małopolskie Voivodeship will be the subject of a separate publication.

In the case of overcast nights in Cracow, a significant absolute difference was found between the decrease in the surface brightness of the night sky in the presence of low- and medium-level clouds (20.1 mcd/m$^2$) and in the presence of high-level clouds (1.8 mcd/m$^2$). However, the relative decrease in surface sky brightness slightly depends on the genus of clouds and amounts to approx. 43%, compared to the situation before switching off the street lighting. This value is similar to that found in the case of the cloudless sky (39%).

Therefore, it can be concluded that turning off all street lighting in a large city reduces the amount of light energy radiated into the sky by about 50%, while reducing the surface brightness of the night sky by about 40%, regardless of the state of the atmosphere.

In the case of smaller cities (Dobczyce and Wieliczka), the effect of reducing the surface brightness of the overcast night sky was not observed. It can only be stated that this change was smaller than the amplitude of changes in brightness associated with the variability of cloud cover. It gives the upper limit of reducing the sky brightness to 60%, which does not exclude a value analogous to that in Cracow.

The measurements of the impact of changes in the brightness of the light source, which is the city, on the zenith surface brightness of the sky at measuring points of different distances, allowed the verification of previous theses [73,76,77]. It was found that these changes are still visible at a distance of 24 km from Cracow, which confirmed the previously established size of the city's light island [25].

**Funding:** This work received no external funding.

**Institutional Review Board Statement:** Not applicable.

**Informed Consent Statement:** Not applicable.

**Data Availability Statement:** The SQM measurements data for all measurement points are available at http://lightpollution.pk.edu.pl/Eng/pomiary.php, accessed on 22 April 2021.

**Acknowledgments:** Authors would like to thank Katarzyna Piotrowicz for providing the meteorological data from the Research Station of the Institute of Geography and Spatial Management of the Jagiellonian University in Cracow (BOT measuring point); Sebastian Wypych for the supervision of the SQM measurements in the Botanical Garden in Cracow (BOT measuring point); Jacek Stasik from Cracow Waterworks for the supervision of the SQM measurements in Dobczyce (DOB measuring points); Marcin Filipek for the SQM measurements in Jerzmanowice (JER measuring point); Dominik Pasternak and Edward Siwek for the supervision of the SQM measurements in Niepołomice (NIE measuring points); the employees of the Mt Suhora Astronomical Observatory, especially Waldemar Ogłoza and Marek Dróżdż, for the supervision of the SQM measurements and also for providing the meteorological data and all-sky camera images (SUH measuring point); Wiesław Kaszowski for the supervision of the SQM measurements (WIE and KAS measuring points); Sławomir Stachniewicz for the SQM measurements and meteorological analysis of the nocturnal clouds at Zabierzów (ZAB measuring point) and also Andrzej Kotarba from Space Research Centre of the Polish Academy of Sciences for valuable comments on VIIRS/DNB data interpretation. Thanks for providing full information on the spatial and temporal scope of switching off the street lighting are also due to city and municipal officials: Krzysztof Płaza (Cracow), Ilona Aniończyk-Kępa (Alwernia); Krzysztof Bilski and Jolanta Zubik (Chrzanów); Stanisław Molik and Paulina Mlostek (Czernichów); Tomasz Suś and Krzysztof Baran (Dobczyce); Ewa Baranowska (Krzeszowice); Eugeniusz Kozioł (Liszki); Wojciech Mackiewicz and Rafał Kubas (Skawina); Elżbieta Kłapa (Wieliczka); and Dariusz Hajto (Zabierzów).

**Conflicts of Interest:** The author declares no conflict of interest.

## Appendix A

The statistical data of communes in the Małopolskie Voivodeship for which the research was conducted (31 December 2018).

**Table A1.** The statistical data of analysed communes.

| Statistical Data | Alwernia | Chrzanów | Dobczyce | Cracow | Krzeszowice | Skawina | Wieliczka | Zabierzów |
|---|---|---|---|---|---|---|---|---|
| Area [km$^2$] | 74 | 79 | 66 | 327 | 139 | 100 | 100 | 99 |
| Population | 12,640 | 47,169 | 15,272 | 771,069 | 32,195 | 43,496 | 59,414 | 26,504 |
| Population/km$^2$ | 170 | 594 | 230 | 2359 | 232 | 436 | 596 | 267 |
| Population urban | 3370 | 36,945 | 6434 | 771,069 | 10,051 | 24,362 | 23,395 | 0 |
| Population rural | 9270 | 10,224 | 8838 | 0 | 22,144 | 19,134 | 36,019 | 26,504 |
| Pop. rural/Pop | 73% | 21% | 52% | 0% | 67% | 41% | 49% | 100% |

## Appendix B

**Table A2.** The length of individual road types according to the OSM classification in the researched communes [km]. The types of roads in which lighting, according to the information obtained, was turned off in 2020 are in bold font.

| Type of Road | Alwernia | Chrzanów | Dobczyce | Cracow | Krzeszowice | Skawina | Wieliczka | Zabierzów |
|---|---|---|---|---|---|---|---|---|
| motorway | 11 | 17 | 0 | **44** | 16 | 0 | 10 | 13 |
| primary | 0 | 11 | 0 | **153** | 11 | 15 | **9** | 13 |
| secondary | 13 | 15 | **16** | **108** | 3 | 13 | **19** | 7 |
| tertiary | 35 | **62** | **35** | **256** | **103** | 47 | **95** | 62 |
| unclassified | **12** | **4** | **29** | **96** | 22 | **42** | **53** | 32 |

**Table A2.** *Cont.*

| Type of Road | Alwernia | Chrzanów | Dobczyce | Cracow | Krzeszowice | Skawina | Wieliczka | Zabierzów |
|---|---|---|---|---|---|---|---|---|
| residential | **108** | **149** | 57 | **978** | 163 | 158 | 201 | **160** |
| service | **40** | 113 | **22** | **1141** | **129** | **80** | 90 | 117 |
| pedestrian | **0** | 0 | 0 | **33** | 2 | **2** | **1** | **1** |
| footway | **21** | 152 | 0 | **1662** | 39 | **57** | **41** | **25** |
| cycleway | **4** | 13 | 0 | **174** | 2 | **8** | 4 | **2** |
| living street | — | — | — | **117** | — | — | — | — |
| Together | 243 | 536 | 168 | 4761 | 490 | 423 | 522 | 431 |
| Together switched off | 185 | 216 | 110 | 4612 | 232 | 189 | 221 | 187 |
| % switched off | 76% | 40% | 66% | 97% | 47% | 45% | 42% | 43% |

## Appendix C

GIS road maps of communes of Alwernia, Chrzanów, Dobczyce and Krzeszowice. The colours correspond to the CIE lighting classification: navy blue – motorway (CIE M Class); shades of red – primary, secondary, tertiary, and unclassified (CIE M Class); shades of blue/green – residential, service, cycleway, pedestrian/footway (CIE P Class). Below are the monthly VIIRS/DNB Cloud Mask images of individual communes generated for March 2020, before the street lighting shutdown period.

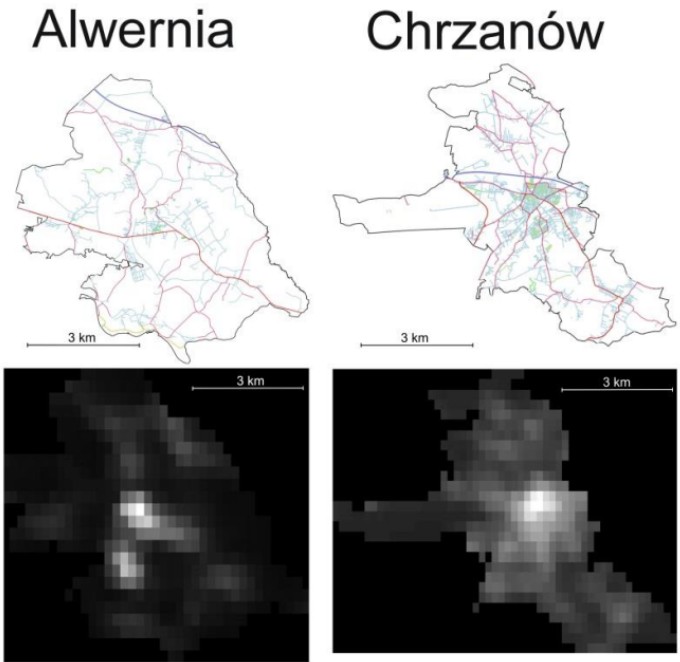

**Figure A1.** *Cont.*

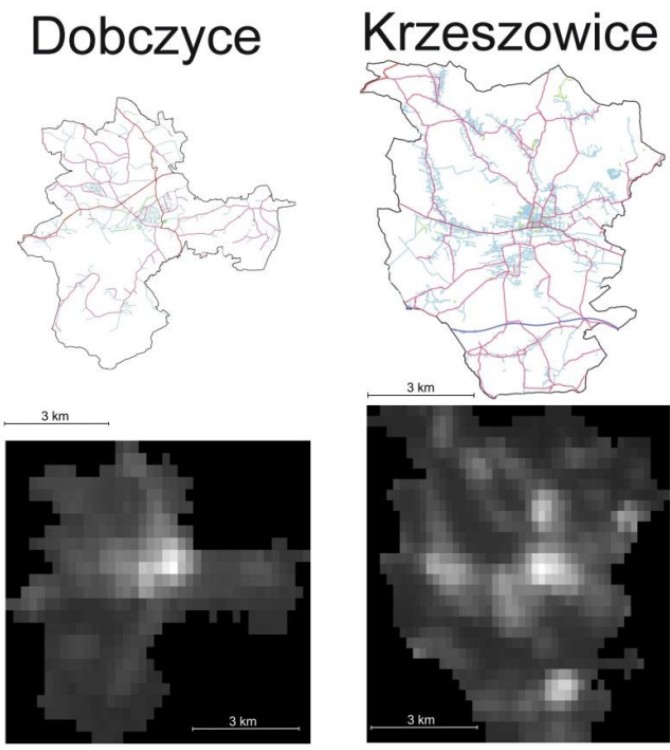

**Figure A1.** Communes of Alwernia, Chrzanów, Dobczyce, and Krzeszowice.

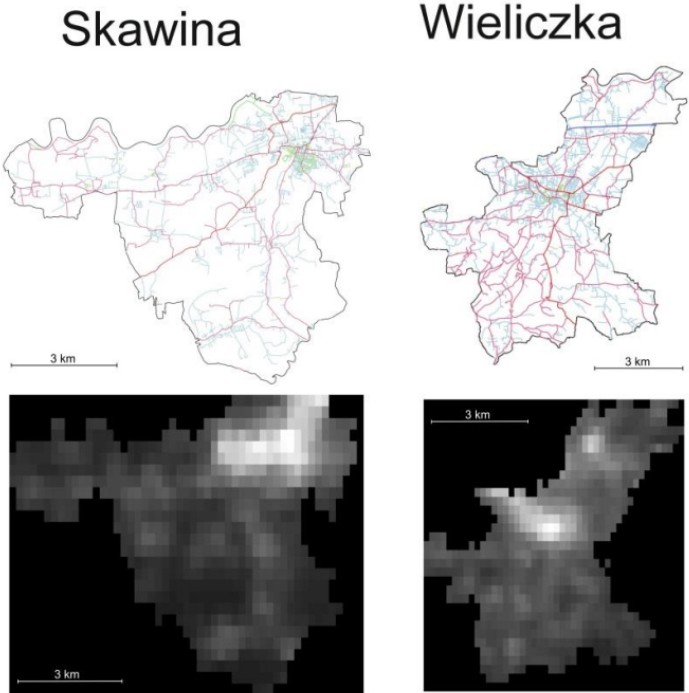

**Figure A2.** *Cont.*

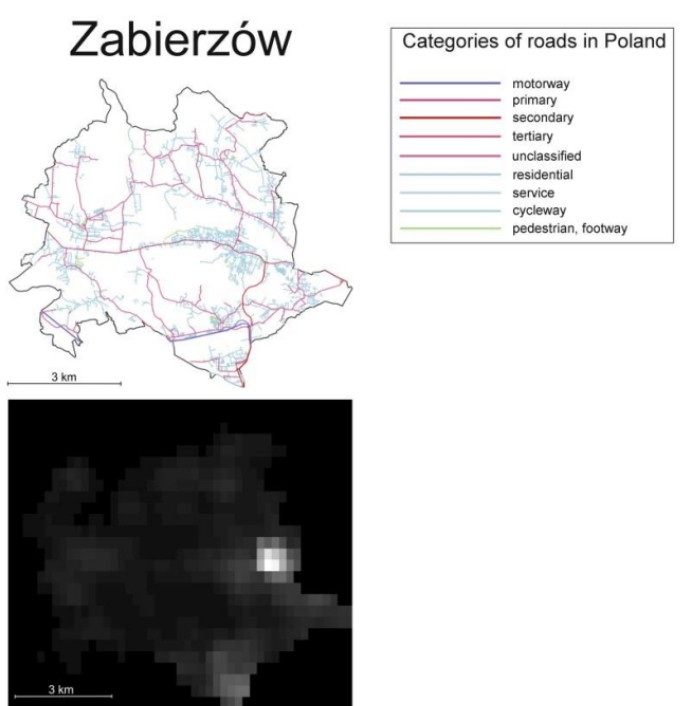

**Figure A2.** Communes of Skawina, Wieliczka, and Zabierzów.

Brightly lit horticultural farm greenhouses dominate in the Zabierzów commune, which makes it difficult or even impossible to make any reliable analysis. The 33,000 m$^2$ farm grew balcony, bedding, and cut flowers (especially roses). The breeding was carried out all year round, also in winter. Currently, the farm is closed since the greenhouses burned down on 10 January 2021.

**Appendix D**

The SQM measuring points (Figure 3; Table 1):

- BOT measuring point (city and commune of Cracow)

The BOT measuring point is located in the very centre of Cracow, within the Field Research Station of the Climatology Department "Botanical Garden" of the Jagiellonian University in Cracow. The SQM-LU-DL meter is mounted on the platform railing located in the middle of the roof of Śniadecki's Collegium of the Jagiellonian University, at an altitude of 17 m above the ground level. The sky brightness at the zenith is measured continuously every 10 min. Access to the data of the meteorological station allows for determining the atmospheric conditions during the measurement. Measurements at this point have been carried out continuously since 2018. However, due to the exhaustion of the batteries and the inability to replace them during pandemic constraints, there was a break in the uninterrupted sequence of measurements from 5 to 25 May 2020.

- DOB measuring point (city and commune of Dobczyce)

The DOB measuring point is located on the outskirts of the Dobczyce city, 23 km south of Cracow, above the drinking water intake on the dam on the Raba River. This point is equipped with an SQM-LE type meter. The sky brightness at the zenith is measured continuously every 10 min. The measuring point has been in operation since 2014, but due to the failure of the computer collecting the data just before the pandemic restrictions were announced and the related difficulties in accessing the meter, measurements on this point did not start again until 5 May 2020.

- JER measuring point (commune Jerzmanowice-Przeginia)

This measuring point is situated in the village of Jerzmanowice in the central part of the Jerzmanowice-Przeginia commune, about 20 km north of Cracow. Measurements are taken on selected nights by a trained amateur astronomer using a handheld SQM-L meter aimed at the zenith. The measurement is performed until three consecutive values are equal with the accuracy of 0.02 mag/arcsec$^2$. In the analysed period, starting from 26 April 2020, two measurements were made on selected nights: the first one approx. 10 min before the moment of switching off street lighting in Cracow and the second one approx. 10 min after that time. This measurement cycle was completed in early May 2020.

- KAS measuring point (commune Mszana Dolna)

This measuring point is located in the area of detached houses in Kasinka Mała, approx. 40 km south of Cracow. The SQM-LU-DL meter is mounted on the roof of the house at a height of approx. 5 m. The brightness of the sky at the zenith is measured continuously every 5 min.

The point was operational from 22 May 2019, with a break from 23 February to 12 April 2020, due to battery exhaustion and pandemic restrictions.

- KPO measuring point (city and commune of Cracow)

The KPO measuring point is situated in the southern part of the Cracow city centre, on an unlit street within a housing estate, approx. 50 m from the nearest apartment blocks. Measurements were made on selected nights by the author of this publication using an SQM-L meter directed at the zenith. The measurement was performed until three successive values were equal with the accuracy of 0.01 mag/arcsec$^2$.

In the analysed period, starting from 15 April 2020, two measurements were made every night: the first one about 15 min before the moment of turning off street lighting in Cracow and the second one about 15 min after that moment. This measurement cycle was completed on 25 May 2020. During the measurements, the weather conditions, in particular the level and genus of cloud cover, were also noted.

- KPO2 measuring point (city and commune of Cracow)

The KPO2 measuring point is situated in the southern part of the centre of Cracow, on the unlit part of allotment gardens, approx. 200 m from the nearest buildings (approx. 400 m from the KPO m. p.). Measurements were made every night from 20 April 2020 by the author of this publication using an SQM-L meter aimed at the zenith. The measurement was performed until three successive values were equal with the accuracy of 0.01 mag/arcsec$^2$. In the analysed period, two measurements were made each night: the first one about 10 min before the moment of switching off street lighting in Cracow and the second one about 10 min after that time. This measurement cycle was completed on 25 May 2020. During the measurements, the weather conditions, in particular the level and genus of cloud cover, were also noted.

- NIE measuring point (city and commune of Niepołomice)

This measuring point is located in the area of detached houses in the northern part of the city of Niepołomice, several kilometres east of Cracow. The SQM-LE meter was originally mounted on the roof of the Młodzieżowe Obserwatorium Astronomiczne (MOA, Youth Astronomical Observatory). Measurements on this point have been carried out continuously since December 2017. Since April 2019, the general reconstruction of the observatory is underway and the measuring point is not functioning. However, due to the possibility of performing unique measurements of sky brightness changes during street lighting shutdowns in neighbouring communes, especially in Cracow, on 7 May 2020, continuous measurement of sky brightness at the zenith started in a place with given coordinates, approx. 900 m from the original location. The meter is situated on the roof of the house, at an altitude of about 10 m, several dozen meters from the nearest street lamps. The sky brightness at the zenith is measured continuously every 5 min. Access to the data of the local meteorological station allows for determining the atmospheric conditions during the measurement.

- SUH measuring point (commune of Niedźwiedź)

The SUH measuring point is situated in the Gorce mountains on top of Mount Suhora, 55 km south of Cracow. The SQM-LE meter is mounted on the terrace of the astronomical observatory of the Pedagogical University of Cracow. The sky brightness at the zenith is measured continuously every 10 min. An independent view from the All-Sky camera and access to the data of the local weather station allows for determining the weather conditions during the measurement. Measurements at this point have been carried out continuously since 2014.

- WIE measuring point (city and commune of Wieliczka)

This measuring point is located in the area of detached houses in the northeastern part of the city of Wieliczka, about 5 km southeast of Cracow. The SQM-LU-DL meter is mounted on the roof of the house, at a height of about 10 m. The brightness of the sky at the zenith is measured continuously every 5 min.

This point is active since 5 May 2019, with a break from 2 March to 9 April 2020, due to battery exhaustion.

- ZAB measuring point (commune of Zabierzów)

The ZAB measuring point is situated in the central part of the Zabierzów commune, about 12 km west from the centre of Cracow, in the area of detached houses. Measurements are made every night by a trained meteorologist using an SQM-L meter aimed at the zenith. Each measurement was performed until three successive values were equal with the accuracy of 0.02 mag/arcsec$^2$. The meteorological data are also recorded, in particular the state and genus of cloud cover. Usually, before the pandemic, such measurement was taken around midnight. In the analysed period, starting from 27 April 2020, two measurements were taken: the first one approx. 10 min before the street lighting was turned off in Cracow and Zabierzów, the second one approx. 10 min after this time. This measurement cycle ended on 25 June 2020.

## Appendix E

Sky Quality Meters (SQMs) are microprocessor-based converters of the frequency signal of the TSL237 brightness sensor, manufactured by Texas Advanced Optoelectronic Solutions Inc. [71] The microprocessor of the SQM has been programmed in a way that the reading from the sensor is converted into the unit of surface brightness ($S_a$), magnitude per square arcsecond (mag/arcsec$^2$), commonly used in astronomy and in the assessment of the quality of the night sky [78]. The TSL237 sensor used in the meters has a frequency output proportional to the irradiance with responsivity 2.3 kHz ($\mu$W/cm$^2$) [79]. However, Unihedron does not reveal what algorithm is used for the conversion of the scale of the sensor ($\mu$W/cm$^2$) to the scale of the meter (mag/arcsec$^2$). Only the relationship between a surface magnitude and a luminance scale is known [67]. The high sensitivity and accuracy of TSL237 in the entire operating range allow for the use of this sensor to measure very small luminous fluxes. Since the spectral sensitivity of the sensor is maximal for the wavelength of ~700 nm, and its characteristics range from ~300 nm to ~1100 nm, i.e., far towards the infrared, the SQM has been additionally equipped with a HOYA CM-500 filter to an additional cut-off of the infrared part of the spectrum. The sensor thus compensated has a maximum sensitivity for a wavelength of ~540 nm, which roughly corresponds to the maximum colour (photopic) vision (555 nm) with a slight shift towards the maximum dark (scotopic) vision (507 nm). This maximum coincides perfectly with the V-band maximum of the Johnes–Cousins system, used in astronomy for photometric measurements corresponding to the sensitivity of the human eye. The full spectral sensitivity curve of the meter ranges from ~320 to ~720 nm, and the relative sensitivity, greater than 10%, ranges from 340 to 680 nm. This is a much broader characteristic than other previously used standards, therefore the Laboratory of Photometry and Radiometry of Light Pollution (LPLAB) of the Institute of Science and Technology of Light Pollution (ISTIL) in Thiene, Italy, where the meter has been thoroughly tested, recommends treating the spectral sensitivity of

SQM as yet another one of the many photometric standards used in astronomy. According to the LPLAB, the differences in the values between SQM and the standard V used in astronomy depend on the type of source being measured, and for the area of application, SQMs range (SQM-V) from $-0.13$ to $+0.11$ mag/arcsec$^2$. For the natural and moderately light-polluted sky, the difference of $0.17 \pm 0.07$ mag/arcsec$^2$ should be taken into account.

In this paper, only the changes in measured night sky brightness are analysed, i.e., the relative values of the indications of a given meter; therefore, both this correction and the one resulting from the glass covering the detector (0.1 mag/arcsec$^2$) are not taken into account.

**Appendix F**

Tables A3–A5 contain the mean annual, monthly, and daily radiances of the analysed communes (except Cracow) [nW/cm$^2$·sr].

**Table A3.** Mean annual radiance for communes in which the street lighting was switched off in 2020 (except Cracow) (SD = 2–6, for Zabierzów: SD = 20–32) [32].

| Year | Alwernia | Chrzanów | Dobczyce | Krzeszowice | Skawina | Wieliczka | Zabierzów |
|------|----------|----------|----------|-------------|---------|-----------|-----------|
| 2012 | 1344 | 3578 | 1064 | 3020 | 2598 | 5220 | 5578 |
| 2013 | 1427 | 3386 | 1168 | 3257 | 2535 | 5921 | 6541 |
| 2014 | 1043 | 3119 | 820 | 2551 | 2553 | 4328 | 6455 |
| 2015 | 1025 | 2933 | 842 | 2538 | 2792 | 4685 | 6173 |
| 2016 | 1220 | 3234 | 908 | 2864 | 2737 | 4850 | 6447 |
| 2017 | 1200 | 3252 | 959 | 2820 | 2968 | 4917 | 6953 |
| 2018 | 1282 | 3318 | 1182 | 3065 | 3153 | 6165 | 6438 |
| 2019 | 1407 | 3621 | 1069 | 3316 | 2995 | 5267 | 6895 |
| 2020 | 1126 | 2843 | 951 | 2736 | 3050 | 4977 | 6857 |

**Table A4.** Mean monthly radiance for communes in which the street lighting was switched off in 2020 (except Cracow). The periods in which the street lighting was turned off for the entire month are in bold font. Italic font indicates that the street lighting was turned off only for a part of the month (SD = 1–6, for Zabierzów: SD = 10–23) [33,54].

| Month | Alwernia | Chrzanów | Dobczyce | Krzeszowice | Skawina | Wieliczka | Zabierzów |
|-------|----------|----------|----------|-------------|---------|-----------|-----------|
| January | 986 | 2383 | 914 | 2748 | 2670 | 4528 | 6453 |
| February | 1242 | 2801 | 1066 | 2783 | 3278 | 5023 | 6133 |
| March | 1464 | 3436 | 1157 | 3283 | 3559 | 5756 | 7019 |
| April | 1355 | 3453 | 1094 | 3372 | 3599 | *3531* | 7216 |
| May | 1313 | 3684 | 1180 | **2544** | *3037* | **2814** | *7114* |
| June | *1194* | *2915* | 1088 | **2416** | **2426** | **2884** | **4605** |
| July | *1031* | **1601** | 1028 | 3142 | **2346** | **2688** | **5288** |
| August | *897* | **1419** | *945* | 2923 | **2429** | **3694** | **5352** |
| September | 1005 | 3113 | *1016* | 3049 | *3282* | 5103 | **6838** |
| October | 972 | 2917 | *867* | *2581* | 2774 | 4613 | **6219** |
| November | 1182 | 3016 | *1031* | 3141 | 3350 | 5128 | **6772** |
| December | **724** | 1948 | *1299* | *1739* | 1894 | 5529 | **4913** |

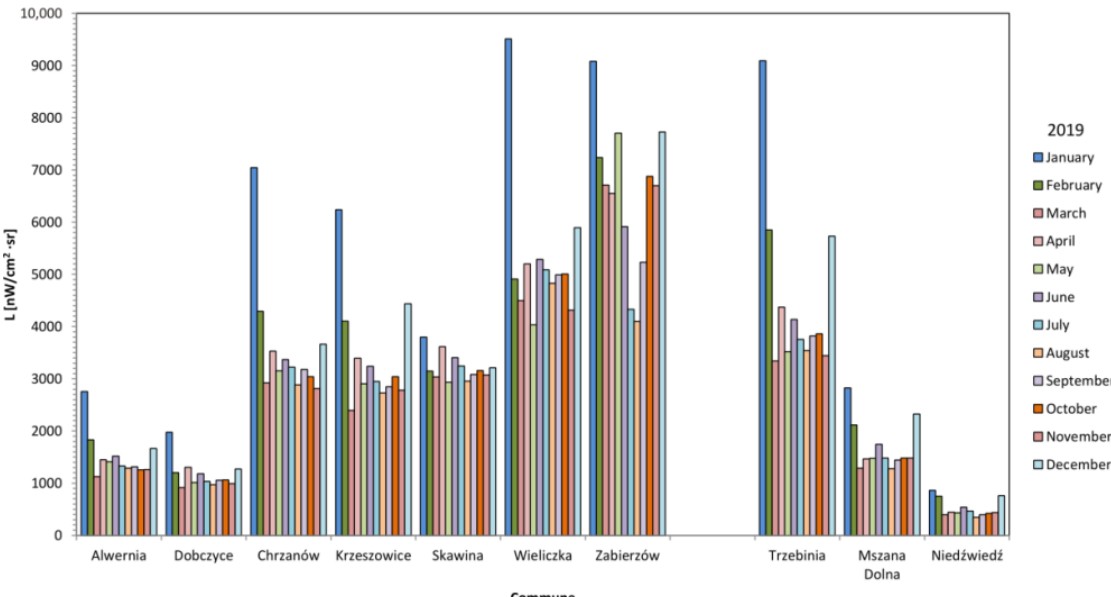

**Figure A3.** Mean monthly radiances in 10 near-Cracow communes in the following months of 2019. Against the background of the practically constant level of radiance throughout the year, high radiance in the winter months, related to the reflection of light from the snow cover, stands out. There is also a noticeable reduction in the radiance in the Zabierzów commune from June to September, related to the reduced lighting of the greenhouse complex in the summer months.

**Table A5.** Examples of the daily radiance in 2020 for the communes of Małopolskie Voivodeship where street lighting was turned off (comm.—the whole commune; urban—the urban part of the commune; rural—the rural part of the commune). Nights are in bold font when, according to the available data, street lighting was turned off completely, in italic font—nights when only a part of the lighting was turned off (SD = 2–6, for Zabierzów: SD = 22–46) [65].

| Date | | Alwernia | Chrzanów | Dobczyce | Krzeszowice | Skawina | Wieliczka | Zabierzów |
|---|---|---|---|---|---|---|---|---|
| | comm. | 1212 | 3310 | 1104 | 3038 | 3530 | 5516 | 6963 |
| 25.03.2020 | urban | 333 | 2382 | 526 | 911 | 2182 | 1982 | – |
| | rural | 858 | 926 | 578 | 2124 | 1348 | 3500 | 6963 |
| | comm. | 1157 | 3101 | 1001 | 3046 | 3350 | 5257 | 7289 |
| 27.03.2020 | urban | 311 | 2210 | 453 | 884 | 2064 | 1912 | – |
| | rural | 831 | 890 | 547 | 2157 | 1286 | 3307 | 7289 |
| | comm. | 1370 | 3746 | 1204 | 3707 | 3818 | 5879 | 7739 |
| 13.04.2020 | urban | 360 | 2699 | 522 | 978 | 2191 | 1926 | – |
| | rural | 986 | 1045 | 682 | 2719 | 1627 | 3928 | 7739 |
| | comm. | 1274 | 3394 | 1079 | 3308 | 3591 | **2511** | 7858 |
| 23.04.2020 | urban | 337 | 2406 | 549 | 888 | 2167 | **913** | – |
| | rural | 911 | 986 | 530 | 2415 | 1424 | **1593** | 7858 |
| | comm. | 1109 | 3186 | 976 | 2994 | 3023 | **2877** | 8524 |
| 27.04.2020 | urban | 314 | 2323 | 408 | 946 | 1570 | **971** | – |
| | rural | 765 | 862 | 568 | 2044 | 1453 | **1899** | 8524 |
| | comm. | 1446 | 4075 | 1453 | *3157* | 3709 | **2990** | 7866 |
| 4.05.2020 | urban | 321 | 2885 | 530 | *624* | 1845 | **833** | – |
| | rural | 1104 | 1184 | 923 | *2528* | 1864 | **2148** | 7866 |

**Table A5.** *Cont.*

| Date | | Alwernia | Chrzanów | Dobczyce | Krzeszowice | Skawina | Wieliczka | Zabierzów |
|------|------|------|------|------|------|------|------|------|
| 21.05.2020 | comm. | *1292* | 4429 | 1179 | *2747* | *3359* | **2893** | *6770* |
| | urban | *372* | 3225 | 579 | *557* | *2207* | **1087** | – |
| | rural | *903* | 1201 | 600 | *2187* | *1152* | **1799** | *6770* |
| 19.07.2020 | comm. | 1411 | **1946** | *938* | *3682* | **2885** | **2019** | *5607* |
| | urban | 298 | **1016** | *355* | *831* | **1684** | **410** | – |
| | rural | 1096 | **927** | *583* | *2840* | **1201** | **1598** | *5607* |
| 20.08.2020 | comm. | 1406 | **2427** | *992* | 4148 | **2694** | **2912** | *4510* |
| | urban | 311 | **1638** | *335* | 938 | **1161** | **902** | – |
| | rural | 1077 | **787** | *656* | 3204 | **1534** | **2000** | *4510* |
| 23.09.2020 | comm. | 762 | *2837* | *897* | 2732 | 3222 | 4995 | *7914* |
| | urban | 254 | *2056* | *470* | 846 | 1966 | 1692 | – |
| | rural | 499 | *778* | *427* | 1881 | 1256 | 3279 | *7914* |
| 23.10.2020 | comm. | 998 | 3269 | *1013* | 3038 | 3291 | 5193 | *7252* |
| | urban | 243 | 2347 | *503* | 807 | 1900 | 1708 | – |
| | rural | 739 | 918 | *509* | 2225 | 1392 | 3456 | *7252* |
| 19.11.2020 | comm. | 905 | 3133 | *1014* | 3036 | 3324 | 5161 | *7458* |
| | urban | 259 | 2267 | *546* | 895 | 2023 | 1895 | – |
| | rural | 635 | 863 | *469* | 2137 | 1301 | 3233 | *7458* |
| 7.12.2020 | comm. | 922 | 2603 | *972* | 2781 | 2943 | 4417 | *9407* |
| | urban | 220 | 1833 | *421* | 906 | 1570 | 1438 | – |
| | rural | 689 | 766 | *551* | 1871 | 1373 | 2956 | *9407* |
| 12.01.2021 | comm. | 999 | 3630 | *1089* | 4147 | 3855 | 5468 | 9361 |
| | urban | 243 | 2520 | *583* | 1004 | 2367 | 1657 | – |
| | rural | 734 | 1107 | *505* | 3137 | 1489 | 3786 | 9361 |

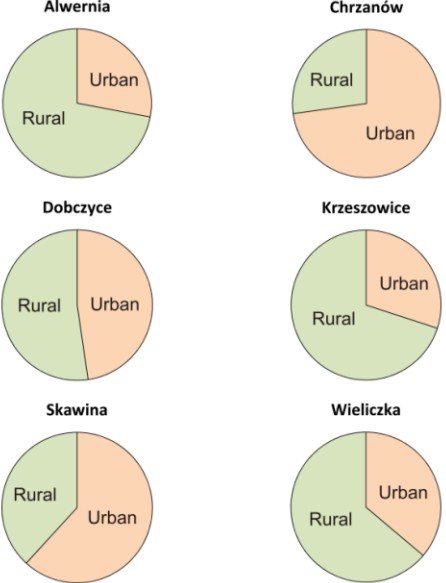

**Figure A4.** Share of urban and rural components in the total radiance of a given commune in the period before switching off street lighting.

**Appendix G**

The tables contain data from SQM measurements made in Cracow and the neighbouring communes in clear sky conditions (Table A6, Table A7) and the cloudy ones (Table A8). Only these nights were taken into account when no changes in the genera of clouds or fog were found during the measurement. The values measured about 10 min before turning off the street lighting in a given commune or the neighbouring one were marked as $S_{a1}$, and measurements made about 10 min after this event were marked as $S_{a2}$ (in the case of April measurements at the BOT measuring point, the values measured before dawn were also taken into account, when the lighting was turned on at 4:00 CEST). In the case of the WIE measuring point, where the lighting was initially turned off in two stages (at midnight and 1:00 CEST), the total brightness decrease was given. The last column of Table A6 shows the suggested source of the $S_a$ changes at a given measuring point.

**Table A6.** Examples of changes of $S_a$ ($dS_a$) determined in the conditions of a cloudless and moonless night at few measurement points (the suggested source of changes is given in brackets). Values are given in mag/arcsec$^2$, in parentheses in mcd/m$^2$.

| Measuring Point | Date | $S_{a1}$ | $S_{a2}$ | $dS_a$ | Notes |
|---|---|---|---|---|---|
| BOT | 2020-04-15 | 18.34 (5.03) | 18.86 (3.11) | 0.52 (1.91) | Cracow |
| | 2020-04-16 * | 18.97 (2.81) | 18.45 (4.54) | 0.52 (1.73) | Cracow |
| | 2020-04-17 * | 19.12 (2.45) | 18.58 (4.03) | 0.54 (1.58) | Cracow |
| | 2020-04-18 | 18.46 (4.50) | 19.00 (2.74) | 0.54 (1.76) | Cracow |
| | 2020-04-19 | 18.61 (3.92) | 19.13 (2.43) | 0.52 (1.49) | Cracow |
| | 2020-04-20 | 18.47 (4.46) | 19.00 (2.74) | 0.53 (1.72) | Cracow |
| | 2020-04-21 | 18.52 (4.26) | 19.05 (2.61) | 0.53 (1.64) | Cracow |
| | 2020-04-22 | 18.51 (4.30) | 19.04 (2.64) | 0.53 (1.66) | Cracow |
| | 2020-04-23 | 18.47 (4.46) | 19.00 (2.74) | 0.53 (1.72) | Cracow |
| | 2020-04-28 | 18.35 (4.98) | 18.90 (3.00) | 0.55 (1.98) | Cracow |
| DOB | 2020-05-17 | 21.35 (0.31) | 21.49 (0.28) | 0.14 (0.04) | Wieliczka |
| | 2020-06-20 | 19.38 (1.93) | 19.66 (1.49) | 0.28 (0.44) | Dobczyce |
| | 2020-06-23 | 19.54 (1.66) | 19.81 (1.30) | 0.27 (0.36) | Dobczyce |
| JER | 2020-04-27 | 20.83 (0.51) | 20.91 (0.47) | 0.08 (0.04) | Cracow |
| | 2020-05-16 | 20.01 (1.08) | 20.09 (1.00) | 0.08 (0.08) | Cracow |
| NIE | 2020-05-17 | 19.56 (1.63) | 19.64 (1.52) | 0.08 (0.11) | Cracow |
| | 2020-05-21 | 19.88 (1.22) | 19.92 (1.17) | 0.04 (0.05) | Cracow |
| | 2020-05-22 | 19.94 (1.15) | 20.00 (1.09) | 0.06 (0.06) | Cracow |
| KPO | 2020-04-16 | 18.49 (4.38) | 19.00 (2.74) | 0.51 (1.64) | Cracow |
| | 2020-04-19 | 18.62 (3.88) | 19.14 (2.41) | 0.52 (1.48) | Cracow |
| | 2020-04-20 | 18.50 (4.34) | 19.00 (2.74) | 0.50 (1.60) | Cracow |
| | 2020-04-21 | 18.59 (3.99) | 18.99 (2.76) | 0.40 (1.23) | Cracow |
| | 2020-04-22 | 18.42 (4.67) | 18.97 (2.81) | 0.55 (1.86) | Cracow |
| | 2020-04-26 | 18.55 (4.14) | 19.07 (2.57) | 0.52 (1.58) | Cracow |
| | 2020-04-28 | 18.44 (4.58) | 18.94 (2.89) | 0.50 (1.69) | Cracow |
| | 2020-05-12 | 18.57 (4.07) | 19.14 (2.41) | 0.57 (1.66) | Cracow |
| | 2020-05-21 | 18.75 (3.44) | 19.24 (2.19) | 0.49 (1.25) | Cracow |

**Table A6.** *Cont.*

| Measuring Point | Date | $S_{a1}$ | $S_{a2}$ | $dS_a$ | Notes |
|---|---|---|---|---|---|
| KPO2 | 2020-04-20 | 18.50 (4.34) | 19.10 (2.50) | 0.60 (1.84) | Cracow |
| | 2020-04-21 | 18.59 (3.99) | 19.17 (2.34) | 0.58 (1.65) | Cracow |
| | 2020-04-22 | 18.49 (4.38) | 19.05 (2.61) | 0.56 (1.76) | Cracow |
| | 2020-04-23 | 18.49 (4.38) | 19.07 (2.57) | 0.58 (1.81) | Cracow |
| | 2020-04-27 | 18.45 (4.54) | 19.02 (2.69) | 0.57 (1.85) | Cracow |
| | 2020-04-28 | 18.44 (4.58) | 18.94 (2.89) | 0.50 (1.69) | Cracow |
| | 2020-05-15 | 18.38 (4.84) | 19.00 (2.74) | 0.62 (2.11) | Cracow |
| | 2020-05-21 | 18.81 (3.26) | 19.26 (2.15) | 0.45 (1.11) | Cracow |
| WIE | 2020-04-15 | 19.99 (1.10) | 20.18 (0.92) | 0.19 (0.18) | Cracow |
| | 2020-04-16 | 20.00 (1.09) | 20.46 (0.71) | 0.46 (0.38) | Cracow + Wieliczka |
| | 2020-04-19 | 20.13 (0.97) | 20.61 (0.62) | 0.48 (0.35) | Cracow + Wieliczka |
| | 2020-04-20 | 20.08 (1.01) | 20.57 (0.64) | 0.49 (0.37) | Cracow + Wieliczka |
| | 2020-04-22 | 20.06 (1.03) | 20.60 (0.63) | 0.54 (0.40) | Cracow + Wieliczka |
| | 2020-05-16 | 20.00 (1.09) | 20.47 (0.71) | 0.47 (0.38) | Cracow + Wieliczka |
| | 2020-05-18 | 20.04 (1.05) | 20.55 (0.66) | 0.51 (0.39) | Cracow + Wieliczka |
| | 2020-05-20 | 20.16 (0.94) | 20.61 (0.62) | 0.45 (0.32) | Cracow + Wieliczka |
| | 2020-05-21 | 20.18 (0.92) | 20.69 (0.58) | 0.51 (0.34) | Cracow + Wieliczka |
| | 2020-06-13 | 19.97 (1.12) | 20.15 (0.95) | 0.18 (0.17) | Wieliczka |
| ZAB | 2020-04-28 | 20.04 (1.05) | 20.19 (0.91) | 0.15 (0.14) | Cracow |

*—The morning measurements at 4:00 CEST.

**Table A7.** Mean change of $S_a$ at measuring points ($dS_a$). The commune that is the suggested source of the change is given in parentheses.

| Measuring Point | $dS_a[mag/arcsec^2]$ | $dS_a[mcd/m^2]$ |
|---|---|---|
| BOT (only Cracow) | $0.53 \pm 0.01$ | $1.72 \pm 0.14$ |
| DOB_1 (Dobczyce) | $0.28 \pm 0.01$ | $0.40 \pm 0.04$ |
| DOB_2 (Wieliczka) | $0.14 \pm 0.05$ | $0.04 \pm 0.02$ |
| JER (Cracow) | $0.08 \pm 0.01$ | $0.06 \pm 0.02$ |
| NIE_1 (Cracow) | $0.06 \pm 0.02$ | $0.07 \pm 0.02$ |
| NIE_2 (Wieliczka) | $0.03 \pm 0.01$ | $0.03 \pm 0.03$ |
| KPO (Cracow) | $0.51 \pm 0.04$ | $1.55 \pm 0.19$ |
| KPO2 (Cracow) | $0.56 \pm 0.05$ | $1.73 \pm 0.27$ |
| WIE_1 (Cracow) | $0.25 \pm 0.02$ | $0.17 \pm 0.06$ |
| WIE_2 (Wieliczka) | $0.23 \pm 0.01$ | $0.16 \pm 0.01$ |
| WIE_3 (Wieliczka + Cracow) | $0.49 \pm 0.03$ | $0.37 \pm 0.03$ |
| ZAB (Cracow) | $0.15 \pm 0.02$ | $0.14 \pm 0.04$ |

**Table A8.** Changes of $S_a$ recorded at measuring points KPO, KPO2, and ZAB during cloudy nights, related to turning off the street lighting in Cracow [mag/arcsec$^2$, in brackets mcd/m$^2$] (data on the height of clouds at the time of measurement come from the meteorological reports of the Kraków-Balice Airport, located on the western border of the city [80]).

| Measuring Point | Date | $S_{a1}$ | $S_{a2}$ | $dS_a$ | Level of Clouds |
|---|---|---|---|---|---|
| KPO | 2020-04-24 | 18.42 (4.67) | 19.03 (2.66) | 0.61 (2.01) | High-level clouds |
| | 2020-04-25 | 15.53 (66.86) | 16.20 (36.07) | 0.67 (30.79) | Medium-level clouds (1400 m) |
| | 2020-04-26 | 16.23 (35.09) | 16.93 (18.41) | 0.70 (16.67) | Medium-level clouds (1829 m) |
| | 2020-04-28 | 18.34 (5.03) | 18.99 (2.76) | 0.65 (2.26) | High-level clouds |
| | 2020-04-29 | 18.44 (4.58) | 18.94 (2.89) | 0.50 (1.69) | High-level clouds |
| | 2020-04-30 | 15.60 (62.68) | 16.10 (39.55) | 0.50 (23.13) | Low-level clouds (274 m) |
| | 2020-05-26 | 15.84 (50.25) | 16.68 (23.18) | 0.84 (27.07) | Medium-level clouds (1829 m) |
| KPO2 | 2020-04-24 | 18.49 (4.38) | 19.07 (2.57) | 0.58 (1.81) | High-level clouds |
| | 2020-04-25 | 15.57 (64.44) | 16.22 (35.41) | 0.65 (29.03) | Medium-level clouds (1400 m) |
| | 2020-04-26 | 16.10 (39.55) | 16.83 (20.19) | 0.73 (19.36) | Medium-level clouds (1829 m) |
| | 2020-04-28 | 18.45 (4.54) | 19.02 (2.69) | 0.57 (1.85) | High-level clouds |
| | 2020-04-29 | 18.44 (4.58) | 18.94 (2.89) | 0.50 (1.69) | High-level clouds |
| | 2020-04-30 | 15.91 (68.73) | 16.72 (43.37) | 0.50 (25.36) | Low-level clouds (274 m) |
| | 2020-05-26 | 15.91 (47.11) | 16.72 (22.34) | 0.81 (24.77) | Medium-level clouds (1829 m) |
| ZAB | 2020-05-27 | 17.57 (9.75) | 17.62 (10.21) | 0.05 (0.46) | Stratus |
| | 2020-05-29 | 18.56 (4.10) | 17.69 (9.14) | 0.87 (5.04) | Stratocumulus (?) |
| | 2020-05-30 | 18.90 (3.00) | 17.96 (7.13) | 0.94 (4.13) | Stratocumulus (?) |
| | 2020-06-01 | 17.56 (10.31) | 17.63 (9.66) | 0.07 (0.64) | Nimbostratus (rain) |

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
