# Peer review of "Effect of Street Lighting on the Urban and Rural Night-Time Radiance and the Brightness of the Night Sky"

_remotesensing, doi:10.3390/rs13091654_

Round 1

Reviewer 1 Report

Dear Author,

First of all let me know there is a very good experiment opportunity these lights shutdown happened during Covid lockdowns. So this experiment is very interesting and can be a contribution to the study of public lighting effects.

I have some comments and suggestions in different sections and specially there are some parts that in my opinion needs to be rewritten or plots redo to give a better performance to this work.

I will go section by section with my comments (I will refer to line and pages of the current manuscript version)

* Abstract

The focus of the abstract is ok but I have some comments:

  • Line 7 . The text “tourist traffic” is not clear for me. What it means? Car traffic? Or it is related to touristic visitors or activities. I think it needs a better English word in any case.
  • Line 12. Now is typed SQM with no definition of the device, so it is unclear. Because later SQM is explained it could be better just say here ground based measurements or in cas to put SQM do it in the full text form.
  • Line 18. Similar to previous one. When VIIRS is referred maybe it is better to talk of satellite data and let VIIRS name for the description done in the main text.

* Introduction

The introduction is good and go step by step introduction so not many comments here and just few recommendations.

  • Line 95. When ISS images are introduced as an option for the measurements. Here for me it is clear to add a citation to the pioneering works using ISS made by A. Sánchez de Miguel (e.g. https://www.sciencedirect.com/science/article/pii/S0034425719300410 )
  • An extra recommendation when COVID-19 is introduced around line 119 or maybe for general approach is the addition of other studies on other branches around “impact” of COVID-19 lockdowns. For example an interesting research on this topic is the one around seismic noise https://science.sciencemag.org/content/369/6509/1338.abstract I think the fact to add that other fields are also detecting effects can give more “power” to this work.
  • Figure 2. I am not sure if this is the right place for this picture. I will comment later in my opinion a more useful place to include it.

* Materials and Methods

I think this section is quite clear and includes almost everything needed to understand the results and discussions sections but I still have some corrections and suggestions specially linked to figures and tables.

  • Figure 4. For me it is mandatory to add an scale or something similar to have an idea of distances in these maps. Now reader have no idea of the size of everything in these panels.
  • Continuing with figure 4. I think the panels are too small. So I strongly recommend to increase the size of this figure to a full page figure. As an alternate option it could be to select the most two relevant villages to show it in bigger size and all the rest can be at big size too but on Appendix sections. I prefer the first option to keep as now but bigger.
  • Line 236. Now there is citation to figure 2, the one with pictures of the change of lighting in the street. So I sincerely recommend to move figure 2 to this position because here is more useful to catch the idea of the effects of this shutdown of lighting.
  • Figure 5. In this section all the measuring points are described, so for me it is mandatory to add these points to the map of this figure. Probably using different symbols for the different methods (permanent, punctual, etc). So later in section 2.1.3 you can refer to figure 5 in more powerful way having the measurement points showed in map.
  • Starting on line 300 up to line 454 there is an extremely long description of the measurement points. I think it is mandatory to redo this in a easy reading way. My proposal is to do a table with the basic information of each station (name, position, kind of site, kind of measurements, comments) If more detailed is planned to included and not fits well to table could be an option to do a basic table here and let a full detail of the measurement sites in Appendix section.
  • On section 2.2.2 when discussing around SQM units. An important and basic information is missed here. There is a wonderful paper analysing this question and needs to be considered here Bara et al (Sensors 2019 https://www.mdpi.com/1424-8220/19/6/1336).
  • On section 2.4 there is a extremely detailed description of Sky Quality Meters device. Almost all the information from line 534 to 578 are not original of this work. So it is absolutely mandatory to cite references used and this could allow to reduce the amount of technical details showed here. I recommend to cite references like Hanel et al (https://www.sciencedirect.com/science/article/abs/pii/S0022407317304442 ) where SQM is described and some references therein. Also ISTIL results are described so some references seems missed, where this information comes from? Maybe some could be in this technical report http://unihedron.com/projects/darksky/sqmreport_v1p4.pdf

* Results and Discussion

Around these sections I think it is mandatory to reorganize it in a better way, so it could be more easy for reader to discover the results showed. Now author exposes the results on sect. 3 and all the important comments on them are appearing on sect. 4, so reader needs to go back to check tables and figures and made the paper unreadable. My proposal is to include the main information from lines 740 to 800 in section 3 accompanying the plots and graph of these results. The reason is that this information requires to check the figures and is more a description of results than a real discussion.

In any case I will now comment the other minor suggestions on these sections:

  • Line 594-Figure 6. When the word variability appears and figure 6 is checked, there are some important variation on L across the period showed. Are there any explanation of these huge changes? Specially the change from 2016 to 2017. I think this figure variations need some explanation if it is possible (change of lighting systems?)
  • On the same way table 2, showed an interesting variation on March-April period across the years. It is like some years March is higher than April and others in the opposite way. Maybe it is just uncertainties of measurements but I do not know if any anthropogenic question can be present there (e.g. Vaccation periods, Easter events,..) This is of course not the aim of your work but if you want to evaluate variations is good to know what happened here.
  • Figure 8. I think these VIIRS data needs to be shown in a bigger way to identify better the changes. Also some references (like roads of figure 4) could be overlapped to identify what is changing.
  • Lines 624-627 and table 3. On these lines some dates are explained as the ones used but in the table more dates are appearing without the explanations. I think text description of dates has to include all the dates showed on table 3. Maybe an option is merge text and table in one table including date, radiance and reason to use it.
  • Lines 638-642. Author exposes some sites as test sites. These ones are almost not discussed and some of them have important changes as seen in figure 10, so I think more information or explanation on test data are mandatory.
  • Figure 11. The bottom panel of Zabierzów. The greenhouse is dominating the image as author explained before. Maybe some mention here is recommended. Later on current section 4 this information appears, so it is clear again that many information of section 4 needs to be included on section 3.
  • Figure 12. Now this figure is very unclear. It is mandatory to add lines or arrows to define the moment of shutdown in each location, similar to what is done on figure 13. If not is not easy to identify the described changes.
  • Going to section 4. The parts linked to figures 16,17,18,… are clearly a good discussion of results. So this has to be the discussion and move the part directly linked to results to sect. 3 as I said before.
  • Lines 864-868. Author explains that clouds effects is linked to previous studies (cite 51). I think it is good to know if this connects with pioneering works on interaction with clouds done by other authors (e.g. Ribas et al 2016 https://doi.org/10.26607/ijsl.v18i0.19 or Jechow et al 2017 https://www.nature.com/articles/s41598-017-06998-z ). Checking if results of clouds studies are in agreement could give more power to cloud conclusions of this work.

* Conclusions

This section is enough clear and the conclusions are consistent with what is described before, so I have no comments on this section.

Reviewer 2 Report

This is an interesting research topic. The presentation is quite good. I have two questions:

  1. Please clarify the novelty of the investigations.
  2. Please clarify the value of the investigations. 

Reviewer 3 Report

The nightly switch off of public lighting provides a good opportunity to estimate the relative effect of street light on sky brightness and the cities' light emission. Thus, the light measurement analysis provides an important addition to the knowledge about the formation of light pollution. The paper deserves publication for these points.

However, before the publication of the paper, I recommend some corrections in the manuscript.

  • Please correct all the data to show only the significant digits. For example, the radiance sums in table 1 are given for 7! digits which are definitely not compatible with the error of the numbers.  please check all the data in the paper
  • The definition of radiometry around lines 461-462 is not precise enough. Radiometry is not restricted to the whole spectrum, and photometry, strictly speaking, is only the measurements with the human eyes' response function. So, strictly speaking, the SQM measurement represents radiometry since the spectral response of SQM is quite different from the V(lambda) function.
  • Conversion from mag/as^2 to cd/m^2 is only an approximation because of the different spectral responses.
  • The author states (line 574) that "As a result, 0.02 mag/arcsec2 was assumed as the 574 measurement accuracy." it is not correct since this only gives the short term repetition error and not the long term variation (e.g. due to temperature changes) or the absolute error. The published error range is rather in the rider of 0.1 mag/as^2.

In my opinion, the paper is somehow lengthy. There are redundancies, as the repetition of a sentence starting with "The spreading of a COVID-19 pandemic in 2020 prompted many ..." The description of the SQM measurements points fits more to the appendix, as it provides details that are not directly necessary for the conclusion. Please consider shortening the main part of the paper.

Reviewer 4 Report

The manuscript deals with a lighting switch-off in the City of Cracow, Poland and some neighboring communes.

Due to the COVID pandemic, many, most and sometimes almost all public street lights were switched off sometimes for months.

This is a unique dataset for light pollution studies that is worth being published.

The author has used satellite night-time ligt data from VIIRS DNB and ground based SQM to study the effect of this switch-off on the upward night-time radiance and the night sky brightness at zenith.

The author used annual, monthly and nightly data for satellites and single point as well as long term continous datafor SQMs.

Overall, the work is an extremely valuable addition to light pollution research.

And despite the many comments I think this is actually very well done as it is.

-----------------------------------------------------------------------------

However, while the overall analysis is indeed mostly well done, there are some points were the presentation, the discussion and the connection to existing work should be improved.

On the other hand a lot of unnecessary information is given (in my view) that - when removed - improves readability.

These are things that can actaully be easily fixed.

------------------------------------------------------------------

The most important point is that the author missed two very important publications that deal with a city wide dimming experiment done in Tucson, Arizona, USA recently:

Kyba, C. C. M., Ruby, A., Kuechly, H. U., Kinzey, B., Miller, N., Sanders, J., ... & Espey, B. (2020). Direct measurement of the contribution of street lighting to satellite observations of nighttime light emissions from urban areas. Lighting Research & Technology, 1477153520958463.

Barentine, J. C., Kundracik, F., Kocifaj, M., Sanders, J. C., Esquerdo, G. A., Dalton, A. M., ... & Kyba, C. C. (2020). Recovering the city street lighting fraction from skyglow measurements in a large-scale municipal dimming experiment. Journal of Quantitative Spectroscopy and Radiative Transfer, 253, 107120.

In that work, the authors observed something similar what is presented here, but with a dedicated dimming experiment and not a pandemic-induced switch off.

So the data is very different, but the and but need to be discussed.

Also in that work, the authors draw a very different conclusion, I think mainly because Tucson has a very well designed "dark sky friendly" street light system and Cracow seem to not have such a system, yet.

-------------------------------------------------------------------

Then there are other switch-off night sky brightness datasets:

During a light pollution campaign ornamental lights in Balaguer, Spain and street lights in a small village neighboring Montsec Astronomical Parc were switched off:

Jechow, A., Ribas, S. J., Domingo, R. C., Hölker, F., Kolláth, Z., & Kyba, C. C. (2018). Tracking the dynamics of skyglow with differential photometry using a digital camera with fisheye lens. Journal of Quantitative Spectroscopy and Radiative Transfer, 209, 212-223.

And the annual switch off during WWF Earth Hour was analyzed in Berlin, Germany:

Jechow, A. (2019). Observing the impact of WWF Earth Hour on urban light pollution: a case study in Berlin 2018 using differential photometry. Sustainability, 11(3), 750.

--------------------------------------------------------------------------------

Then, I am missing a proper discussion about two other studies on COVID pandemic that have been actually cited as ref. 30 and 31.

In both of those studies NO switch off was observed, but still the NSB and the satellite night-time data changed to some extent.

Both studies come to the conclusion that reduced air pollution and reduced scattering is a cause for improved night skies.

See particularly ref. 31 on air traffic.

This is actually a field that was pioneered by the author of the present manuscript and can be easily commented on - also because the author must know Cracow and air traffic very well.

--------------------------------------------

Minor comments include to improve graphics and add scales to VIIRS data, add potentially some information and improve readability.

--------------------------------------------

I am looking forward to the revised version

Detailed comments here:

---------------------------------------------------------
Title: maybe add "night-time" before "radiance"

-------------------------------------------------------

Abstract:

Overall: the abstract could be more concise
- shorten and focus on the important points reduce level of detail of not so importnat information

line 8: mention that Cracow is in Poland
line 9: "city lamps" -> do you mean "public street lights"?
line 11: I actually doubt that this statement is true

see work by Barentine et al.

and 

Kyba et al.

around Tucson

I guess you could state "on a longer time-frame at this scale"

abbreviations need to be introduced and instruments need to be specified better:

SQM - small night-sky radiance meters
VIIRS/DNB data - night-time light satellite data 

line 15: surface brightness - zenith night sky brightness

line 14-17: combine the two sentences "it was found...turning off..15-39%, and up to 20km"

--------------------------------------------------------------------------------------------------------

Introduction:

The work is very exciting but the introduction is rather tyring - it provides a ot of unnecessary information, not up to date references etc.

shorten and get to the exciting work more quickly!

line 29-45: could be shortened 

Ref. 4 is not available but there are many papers around:

I think the first paper explicitely stating skyglow is:

Rosebrugh, D. W., 1935. Sky-glow from large cities. J. Royal Astron. Soc. Can. 29, 79.

and the first to name astronomical light pollution:

Riegel, K.W., 1973. Light pollution: outdoor lighting is a growing threat to astronomy. Science, 179(4080), 1285-1291.

line 45: is this a Polish policy? is there a better citation than ref (5) it is difficult to obtain KZD has many freely available papers

line 46-56: The whole section on modeling seems unnecessary because there is no modeling in the paper

maybe it is worthwhile to mention work by Walker in the context of NTL and population (Fig. 16)

line 57-74: This can be shortened as well drastically

Ref 22 should be updated with the "New World Atlas" based on VIIRS:

Falchi, F., Cinzano, P., Duriscoe, D., Kyba, C. C., Elvidge, C. D., Baugh, K., ... & Furgoni, R. (2016). The new world atlas of artificial night sky brightness. Science advances, 2(6), e1600377.

line 75: also the section on DMSP appears to be obsolete as no DMSP data was used

line 88: "equipped with the best sensors and optical systems in civilian missions so far." - I highly doubt that statement - there are superior optical systems than VIIRS in orbit - day and night - however, I assume you mean something like "...the best freely available night-time dataset with daily and global coverage."

Daytime satellites include hyperspectral sensors like DESIS or PRISMA

During night it is Luojia for example - see recent review:

Levin, N., Kyba, C. C., Zhang, Q., de Miguel, A. S., Román, M. O., Li, X., ... & Elvidge, C. D. (2020). Remote sensing of night lights: A review and an outlook for the future. Remote Sensing of Environment, 237, 111443.

line 97: "communication routes" -> do you mean "commuting routes" or "access routes"?

line 100-118: please mention Kyba et al. Barentine et al. (Tucson) Jechow et al. (Balaguer) and Jechow (Earth Hour)

line 119: please have a look at Ref 30 and 31 again and mention air pollution as a main cause for improved sky brightness (or discuss this later in text)

----------------------------------------------------------------

Material and Methods

I think Fig 3 is not needed. It would be better to have a more detailed map of Malopolskie Voivodship - could be a more detailed Fig 5

line 169-200: Motorway has 2 or 4 lanes? What does dimly lit or pourly lit mean? Are there lighting classes (CIE M, P etc.) available?

line 204: I cannot see the roads easily in the Fig 4 - actually the figure is too small. A suggestion is to pick a larger single Figure and move all the others to Appendix

Also a scale is missing - can this be done in false color?

line 237: Fig. 5 - please add SQM locations

line 317 (417): what is temporal resolution of DOB (SUH) station?

line 386: how was the exact time known?

line 455-489: this subsection reads strange - I recommend deleting it and rather include only the necessary information in the description of VIIRS and SQM, there is now some redundancy - keep it short and mention literature

line 500: this should be nm not µm

-----------------------------------

results

Table 1: is STD given per pixel?

Table 2: please check significant digits - could be highlighted in text that May 2020 was very low

Fig. 6: is it known when lighting technology was changed? it is difficuzlt to draw a big conclusion from this dataset

Fig. 7 maybe in color? (also other similar figures)

Fig. 8 Needs a scale and could be larger - maybe in false-color?

Figure 10: maybe add 2019 data in Appendix? (color?)

taable 4: define rel. dL and dL somewhere

Fig. 11: larger, color, scale

Fig. 12: maybe add legend to figs

Fig. 13: please add that this is clear sky to caption

Fig 14&15: indicate switch-off

-----------------------------------------

Discussion:

Please discuss your result also in context of other key papers on larger switch-offs (Tucson, Berlin, Balaguer) and COVID pandemic (Ref. 30&31)

Also mention Walkers work in context of Fig. 16

line 742: actually the claim of reduced annula radiance seems too far fethced when looking at Fig. 6

In the discusssion of night-time upward rainacne it would be nice to distinguish between sum of lights and per pixel (area) values. A SOL of Cracow will always be higher because of larger area, even if the per pixel values are very different. 

line 817: In the case of Cracow...this resulted in a relatively small decrease in radiance.

This is interesting and does not come out of the abstract, introduction and conclusion. Needs to be highlighted there as well.

It also goes hand in hand with the results of kyba et al. and Barentine et al. in Tucson - should be discussed.

---------------------------------------

Conclusion:

line 885-899: I am missing the Cracow data here (in line 817 it was stated that that was lower- how much needs to go here)

line 906: the PM values and NSB data could be discussed in context to Ref. 31

Round 2

Reviewer 1 Report

Dear authors,

I think the manuscript is clearly much better now with all your updates coming from the peer-review process.

For my side everything is ok except one point that I think it is still not properly solved. This point is section 2.3, according to cover letter the details were moved to Appendix E. It is true that Appendix E has the details but they still remain on section 2.3, so information is appearing two times in this manuscript version. My proposal is to summarize 2.3 to the highlights on SQM and let the details on Appendix E.

All the rest is solved.

Author Response

Reviewer (R): I think the manuscript is clearly much better now with all your updates coming from the peer-review process.

Author(A): Thank you very much for all valuable advice.

R: For my side everything is ok except one point that I think it is still not properly solved. This point is section 2.3, according to cover letter the details were moved to Appendix E. It is true that Appendix E has the details but they still remain on section 2.3, so information is appearing two times in this manuscript version. My proposal is to summarize 2.3 to the highlights on SQM and let the details on Appendix E.

A: Of course, you are right. In section 2.3 I have only left some remarks related to the SQM measurements in this experiment (lines 401-423). The entire description of the SQM operating principle can now be found in Appendix E (lines 996-1030).

R: All the rest is solved.

A: Thank you very much.

Reviewer 2 Report

The author has solved my considerations. 

Author Response

Reviewer: The author has solved my considerations.

Author: Thank you very much for all your valuable advice.

Reviewer 4 Report

the paper has improved a lot

great work

still one last comment:

please add scale to VIIRS data and difference

Fig. 7, 10, C1, C2

Author Response

Reviewer (R): the paper has improved a lot

Author (A): Thank you very much for all your valuable advice.

R: great work

A: It's nice to read such a review

R: still one last comment: please add scale to VIIRS data and difference Fig. 7, 10, C1, C2

A: I wonder why the Reviewer did not notice the scales that were given in all the figures? I came to two conclusions:

  1. Perhaps the scale lines were too narrow and the font size too small?
  2. Perhaps it was not clear that scales given in the road maps (Figures C1, C2) also applied to the VIIRS data below?

I solved these problems in the following ways (I hope):

Ad.1.

I bold the scale lines and made the font bigger (lines 246,470,542,544,877 and 879).

Ad.2.

Regardless of scales given in the road maps, I also added them in all VIIRS data (additionally, also in Fig. 3 (line 246)).

I also changed the layout of Fig.7 - I think it is more legible now (line 470).

In Fig.10, I just bold the scale lines and enlarged the font - I think, that it is clear that the scales to the right of VIIRS images apply to each image in the row (lines 542 and 544).